# Score Models for Offline Goal-Conditioned Reinforcement Learning

**Harshit Sikchi**[*θ] , **Rohan Chitnis**[†φ]**, Ahmed Touati**[†φ]**, Alborz Geramifard**[φ]**, Amy Zhang**[θ,φ]**, Scott Niekum**[ψ]

[θ] University of Texas at Austin, [φ] Meta AI, [ψ] UMass Amherst

## Abstract

Offline Goal-Conditioned Reinforcement Learning (GCRL) is tasked with learning to achieve multiple goals in an environment purely from offline datasets using sparse reward functions. Offline GCRL is pivotal for developing generalist agents capable of leveraging pre-existing datasets to learn diverse and reusable skills without hand-engineering reward functions. However, contemporary approaches to GCRL based on supervised learning and contrastive learning are often suboptimal in the offline setting. An alternative perspective on GCRL optimizes for occupancy matching, but necessitates learning a discriminator, which subsequently serves as a pseudo-reward for downstream RL. Inaccuracies in the learned discriminator can cascade, negatively influencing the resulting policy. We present a novel approach to GCRL under a new lens of mixture-distribution matching, leading to our discriminator-free method: SMORe. The key insight is combining the occupancy matching perspective of GCRL with a convex dual formulation to derive a learning objective that can better leverage suboptimal offline data. SMORe learns *scores* or unnormalized densities representing the importance of taking an action at a state for reaching a particular goal. SMORe is principled and our extensive experiments on the fully offline GCRL benchmark composed of robot manipulation and locomotion tasks, including high-dimensional observations, show that SMORe can outperform state-of-the-art baselines by a significant margin.

## 1 Introduction

A generalist agent will require a vast repertoire of skills, and large amounts of offline pre-collected data offer a way to learn useful skills without any environmental interaction. Many subfields of machine learning like vision and NLP have enjoyed great success by designing objectives to learn a general model from large and diverse datasets. In robot learning, offline interaction data has become more prominent in the recent past (Ebert et al., 2021), with the scale of the datasets growing consistently (Walke et al., 2023; Padalkar et al., 2023). Goal-conditioned reinforcement learning (GCRL) offers a principled way to acquire a variety of useful skills without the prohibitively difficult process of hand-engineering reward functions. In GCRL, the agent learns a policy to accomplish a variety of goals in the environment. The rewards are sparse and goal-conditioned: 1 when the agent's state is proximal to the goal and 0 otherwise. However, the benefit of not requiring the designer to hand-engineer dense reward functions can also be a curse, because learning from sparse rewards is difficult. Driving progress in fundamental offline GCRL algorithms thus becomes an important aspect of moving towards performant generalist agents whose skills scale with data.

Despite recent progress in developing methods for goal-reaching in the online setting (where environment interactions are allowed), a number of these methods are either suboptimal in the offline setting or suffer from learning difficulties. Prior GCRL algorithms can largely be classified into one of three categories: iterated behavior cloning, RL with sparse rewards, and contrastive learning. Iterated behavior cloning or goal-conditioned supervised learning approaches (Ghosh et al., 2019; Yang et al., 2019) have been shown to be provably suboptimal (Eysenbach et al., 2022a) for GCRL. Modifying single-task RL methods (Silver et al., 2014; Kostrikov et al., 2021) for GCRL with 0-1 reward implies learning a $Q$-function that predicts the discounted probability of goal reaching, which

---

*Work done partially during an internship at Meta AI. Correspondence to hsikchi@utexas.edu

†Equal Contribution. **Project page (Code and Videos):** hari-sikchi.github.io/smore/

makes it essentially a density model. Modeling density directly is a hard problem, an insight which has prompted the development of methods (Eysenbach et al., 2020) that learn density-ratio instead of densities, as classification is an easier problem than density estimation. Contrastive RL approaches to GCRL (Eysenbach et al., 2020; 2022b; Zheng et al., 2023) aim to do precisely this and are the main methods to enjoy success for applying GCRL in high-dimensional observation spaces. However, when dealing with offline datasets, contrastive RL approaches (Eysenbach et al., 2022b; Zheng et al., 2023) are suboptimal, as they only learn a policy that is a greedy improvement over the Q-function of the data generation policy. This begs the question: *How can we derive a performant GCRL method that learns near-optimal policies from offline datasets of suboptimal quality?*

In this work, we leverage the underexplored insight of formulating GCRL as an occupancy matching problem. Occupancy matching between the joint state-action-goal visitation distribution induced by the current policy and the distribution over state-actions that transition to goals can be shown to be equivalent to optimizing a max-entropy GCRL objective. Occupancy matching has been studied extensively in imitation learning (Ghasemipour et al., 2020) and often requires learning a discriminator and using the learned discriminator for downstream policy learning through RL. Indeed, a prior GCRL work (Ma et al., 2022) explores a similar insight. Unfortunately, errors in learned discriminators can compound and adversely affect the learned policy's performance, especially in the offline setting where these errors cannot be corrected with further interaction with the environment.

Going beyond the shortcomings of the previous methods, our proposed method combines the insight of formulating GCRL as an occupancy matching problem along with an efficient, discriminator-free dual formulation that learns from offline data. The resulting algorithm `SMORe` forgoes learning density functions or classifiers, but instead learns unnormalized densities or *scores* that allow it to produce near-optimal goal-reaching policies. The scores are learned via a Bellman-regularized contrastive procedure that makes our method a desirable candidate for GCRL with high-dimensional observations, avoiding the need for density modeling. Our experiments represent a wide variety of goal-reaching environments – consisting of robotic arms, anthropomorphic hands, and locomotion environments. We lay out the following contributions: 1) on the extended offline GCRL benchmark, our results demonstrate that `SMORe` significantly outperforms prior methods in the offline GCRL setting. 2) In line with our hypothesis, discriminator-free training makes `SMORe` particularly robust to decreasing goal-coverage in the offline dataset, a property we demonstrate in the experiments. 3) We test `SMORe` for zero-shot GCRL on a prior benchmark (Zheng et al., 2023) for high dimensional vision-based GCRL where contrastive RL approaches are the only class of GCRL methods that have been successful, and show improved performance over other state-of-the-art baselines.

## 2 PROBLEM FORMULATION

We consider an infinite horizon discounted Markov Decision Process denoted by the tuple $\mathcal{M} = (\mathcal{S}, \mathcal{A}, p, r, \gamma, d_0)$, where $\mathcal{S}$ is the state space, $\mathcal{A}$ is the action space, $p$ is the transition probability function, $r : \mathcal{S} \times \mathcal{A} \rightarrow \mathbb{R}$ is the reward function, $\gamma \in (0, 1)$ is the discount factor, and $d_0$ is the initial state distribution. We constrain ourselves to the goal-conditioned RL setting, where we additionally assume a goal space $\mathcal{G}$ where states in $\mathcal{S}$ are mapped to the goal space using a known mapping: $\phi : \mathcal{S} \rightarrow \mathcal{G}$. The reward function $r(s, a, g)$ in GCRL is sparse and also depends on the goal. A goal conditioned policy $\pi : \mathcal{S} \times \mathcal{G} \rightarrow \Delta(\mathcal{A})$ outputs a distribution over actions in a given state conditioned on a goal. Given a distribution over desired evaluation goals $q^{\texttt{test}}(g)$, the objective of goal-conditioned RL is to find a policy $\pi_g$[1] that maximizes the expected discounted return:

$$J(\pi_g) := \mathbb{E}_{g \sim q^{\texttt{test}}((g)), s_0 \sim d_0, a_t \sim \pi_g} \left[ \sum_{t=0}^{\infty} \gamma^t r(s_t, a_t, g) \right]. \tag{1}$$

We denote by $P^{\pi_g}$ the transition operator induced by the policy $\pi_g$ defined as $P^{\pi_g} S(s, a, g) := \mathbb{E}_{s' \sim p(\cdot|s,a), a' \sim \pi_g(\cdot|s',g)} \left[ S(s', a', g) \right]$, for any *score* function $S : \mathcal{S} \times \mathcal{A} \times \mathcal{G} \rightarrow \mathbb{R}$. We use $d^{\pi}(s, a \mid g)$ to denote the discounted goal-conditioned state-action occupancy distribution of $\pi_g$, i.e $d^{\pi_g}(s, a \mid g) = (1 - \gamma)\pi(a|s, g) \sum_{t=1}^{\infty} [\gamma^t \text{Pr}(s_t = s|\pi_g, d_0)]$. which represents the expected discounted time spent in each state-action pair by the policy $\pi_g$ conditioned on the goal $g$. For complete generality, in GCRL, the distribution of goals the policy is trained on often differs from the test goal distribution. To make this distinction clear we define the training distribution $q^{\texttt{train}}(g)$, a uniform measure over goals

---

[1]We use the subscript g to make the policy's conditioning on g explicit.

we desire to learn to optimally reach during training. We write $d^{\pi_g}(s, a, g) = q^{\texttt{train}}(g)d^{\pi_g}(s, a \mid g)$ as the joint state-action-goal visitation distribution of the policy $\pi_g$ under the training goal distribution. A state-action-goal occupancy distribution must satisfy the *Bellman flow constraint* in order for it to be a valid occupancy[2] distribution for some stationary policy $\pi_g$, $\forall s \in \mathcal{S}, a \in \mathcal{A}, g \in \mathcal{G}$:

$$d(s, a, g) = (1 - \gamma)d_0(s, g)\pi_g(a \mid s, g) + \gamma \sum_{s', a'} p(s \mid s', a')d(s', a', g)\pi_g(a \mid s, g), \qquad (2)$$

where $d_0(s, g) = d_0(s)q^{\texttt{train}}(g)$. Finally, given $d^{\pi_g}$, we can express the learning objective for the GCRL agent under the training goal distribution as $J^{\texttt{train}}(\pi_g) = \frac{1}{1-\gamma}\mathbb{E}_{(s,a,g)\sim d^{\pi_g}}[r(s, a, g)]$.

In this work, we focus on the offline setup where the agent cannot interact with the environment $\mathcal{M}$ and instead has access to a offline dataset of $\mathcal{D} := \{\tau_i\}_{i=1}^N$, where each trajectory $\tau^{(i)} = (s_0^{(i)}, a_0^{(i)}, r_0^{(i)}, s_1^{(i)}, ...; g^{(i)})$ with $s_0^{(i)} \sim d_0$. The trajectories are usually relabelled with the $q^{\texttt{train}}(g)$ during learning. We denote the joint state-action-goal distribution of the offline dataset $\mathcal{D}$ as $\rho(s, a, g)$.

## 3   SCORE-MODELS FOR OFFLINE GOAL CONDITIONED REINFORCEMENT LEARNING

In this section, we introduce our method in two parts: First, we build up the equivalence of the GCRL objective to the occupancy matching problem in Section 3.1, and then we derive a discriminator-free dual objective for solving the occupancy matching problem using off-policy data in Section 3.2. Finally, we present the algorithm for `SMORe` under practical considerations in Section 3.3.

### 3.1   GCRL AS AN OCCUPANCY MATCHING PROBLEM

Define a *goal-transition distribution* $q(s, a, g)$ in a stochastic MDP as $q(s, a, g) \propto q^{\texttt{train}}(g)\mathbb{E}_{s'\sim p(\cdot|s,a)}[\mathbb{I}_{\phi(s')=g}]$. Intuitively, the distribution has probability mass on each transition that leads to a goal. We formulate the GCRL problem as an occupancy matching problem by searching for the policy $\pi_g$ that minimizes the discrepancy between its state-action-goal occupancy distribution and the goal-transition distribution $q(s, a, g)$:

$$\texttt{Occupancy matching problem:} \quad \mathcal{D}_f(d^{\pi_g}(s, a, g)\|q(s, a, g)), \qquad (3)$$

where $D_f$ denotes an $f$-divergence with generator function $f$. Note that the $q$ distribution is potentially unachievable by any goal-conditioned policy $\pi_g$. Firstly, it does not account for the initial transient phase that the policy must navigate to reach the desired goal. Secondly, even if we consider only the stationary regime (when $\gamma \to 1$), it may not be dynamically possible for the policy to continuously remain at the goal and rather necessitate cycling around the goal. However, in Proposition 1, we show that the occupancy matching in Eq. 3 offers a principled objective since it forms a lower bound to the max-entropy GCRL problem.

**Proposition 1.** *Consider a stochastic MDP, a stochastic policy $\pi$, and a sparse reward function $r(s, g) = \mathbb{E}_{s'\sim p(\cdot|s,a)}[\mathbb{I}(\phi(s') = g, q^{\texttt{train}}(g) > 0)]$ where $\mathbb{I}$ is an indicator function. Define a soft goal transition distribution to be $q(s, a, g) \propto exp(\alpha\, r(s, a, g))$. The following bounds hold for any $f$-divergence that upper bounds KL-divergence (eg. $\chi^2$, Jensen-Shannon):*

$$J^{train}(\pi_g) + \frac{1}{\alpha}\mathcal{H}(d^{\pi_g}) \geqslant -\frac{1}{\alpha}\mathcal{D}_f(d^{\pi_g}(s, a, g)\|q(s, a, g)) + C, \qquad (4)$$

*where $\mathcal{H}$ denotes the entropy, $\alpha$ is a temperature parameter and $C$ is the partition function for $e^{R(s,a,g)}$. Furthermore, the bound is tight when $f$ is the KL-divergence.*

Ma et al. (2022) (in Proposition 4.1) presented a similar result connecting state-goal distribution matching ( i.e $D_{KL}(d^\pi(s, g)\|q(s, g))$) to GCRL objective and Proposition 1 extends their results to goal-transition distribution matching. Matching action-free distributions necessitates constructing a loose lower bound that is tractable to optimize. By considering goal-transition distributions we sidestep constructing a loose lower bound and instead directly obtain a tractable distribution matching objective (Ghasemipour et al., 2020; Kostrikov et al., 2019) that is tight under KL-divergence.

How does converting a GCRL objective to an imitation learning objective make learning easier? Estimating the $f$-divergence still requires estimating the joint policy visitation probabilities $d^{\pi_g}(s, a, g)$, which itself presents a challenging problem. We show in the following section that we can leverage

---

[2]We will use "occupancy" and "visitation" interchangeably.

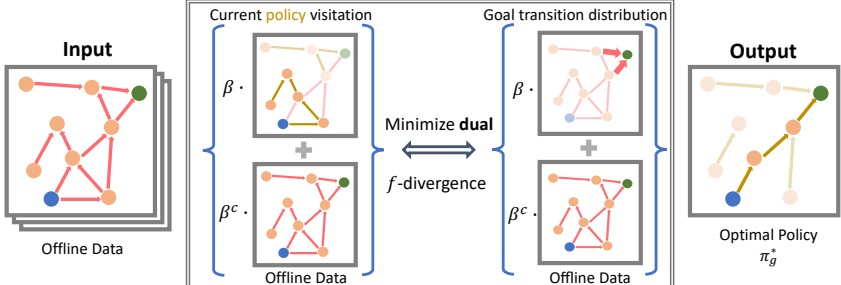

Figure 1: Illustration of the SMORe objective where $\beta^c = 1 - \beta$: SMORe matches a mixture distribution of current policy and offline data to a mixture of the goal-transition distribution and offline data in order to find the optimal goal reaching policy.

convex duality to transform the imitation learning problem into an off-policy optimization problem, removing the need to sample from $d^{\pi_g}(s, a, g)$ whilst being able to leverage offline data collected from arbitrary sources.

## 3.2 SMORe: A Dual Formulation for Occupancy Matching

The previous section establishes GCRL as an occupancy matching problem (Eq. 3) but provides no way to use offline data whose joint visitation distribution is given by $\rho(s, a, g)$. To leverage offline data to learn performant goal-reaching policies, we consider a surrogate objective to the occupancy matching learning problem by matching *mixture* distributions:

$$\min_{\pi_g} \mathcal{D}_f(\text{Mix}_\beta(d^{\pi_g}, \rho) \| \text{Mix}_\beta(q, \rho)), \tag{5}$$

where for any two distributions $\mu_1$ and $\mu_2$, $\text{Mix}_\beta(\mu_1, \mu_2)$ denotes the mixture distribution with coefficient $\beta \in (0, 1]$ defined as $\text{Mix}_\beta(\mu_1, \mu_2) = \beta\mu_1 + (1 - \beta)\mu_2$. Proposition 2 (in appendix) shows the matching mixture distribution[3] provably maximizes a lower bound to the Lagrangian relaxation of the max-entropy GCRL objective subject to a dataset regularization constraint. We can rewrite the mixture occupancy matching objective as a convex program with linear constraints (Manne, 1960; Nachum and Dai, 2020):

$$\max_{\pi_g, d} -\mathcal{D}_f(\text{Mix}_\beta(d, \rho) \| \text{Mix}_\beta(q, \rho))$$

$$\text{s.t } d(s, a, g) = (1 - \gamma)d_0(s, g)\pi(a|s) + \gamma \sum_{s' \in \mathcal{S}} d(s', a', g)p(s|s', a')\pi(a'|s', g), \ \forall s \in \mathcal{S}. \tag{6}$$

An illustration of this objective can be found in Figure 1. Effectively, we have simply rewritten Eq. 5 into an equivalent problem by considering an arbitrary probability distribution $d(s, a, g)$ in the optimization objective, only to later constrain it to be a valid probability distribution induced by some policy $\pi_g$ using the *Bellman-flow constraints*. The motivation behind this construction of the primal form is that we have made computing the Lagrangian-dual easier as this objective is convex with linear constraints. Theorem 1 shows that we can leverage tools from convex duality to obtain an unconstrained dual problem that does not require computing $d^{\pi_g}(s, a, g)$ or sampling from it, while effectively leveraging offline data.

**Theorem 1.** *The dual problem to the primal occupancy matching objective (Equation 6) is given by:*

$$\max_{\pi_g} \min_S \beta(1 - \gamma)\mathbb{E}_{d_0, \pi_g}[S(s, a, g)] + \mathbb{E}_{\text{Mix}_\beta(q, \rho)}[f^*(\gamma P^{\pi_g}S(s, a, g) - S(s, a, g))] \tag{7}$$

$$- (1 - \beta)\mathbb{E}_\rho[\gamma P^{\pi_g}S(s, a, g) - S(s, a, g)],$$

*where $f^*$ is conjugate function of $f$ and $S$ is the Lagrange dual variable defined as $S : \mathcal{S} \times \mathcal{A} \times \mathcal{G} \to \mathbb{R}$. Moreover, as strong duality holds from Slater's conditions the primal and dual share the same optimal solution $\pi_g^*$ for any offline transition distribution $\rho$.*

To our knowledge, the closest prior works to our proposed method are GoFAR (Ma et al., 2022) and Dual-RL (Sikchi et al., 2023). GoFAR considers the special case of KL-divergence for the imitation formulation and derives a dual objective that requires learning the density ratio $\frac{\rho(s, g)}{q(s, g)}$ in the

---

[3]Note that Eq. 5 shares the same global optima as the previous occupancy matching objective at $d_g^\pi(s, a, g) = q(s, a, g)$ when $q$ is an achievable visitation under some policy and recovers the original objective in Eq. 3 when $\beta = 1$.

form of a discriminator and using this as a pseudo-reward. This leads to compounding errors in the downstream RL optimization when learning the density ratio is challenging, e.g. in the case of low coverage between $\rho(s, a, g)$ and $q(s, a, g)$. We show this phenomenon experimentally in Section 4.3. Dual-RL (Sikchi et al., 2023) uses convex duality for matching visitation distribution of realizable expert demonstrations and does not deal with the GCRL setting. *Our contribution is a novel method for GCRL that is discriminator-free, applicable for a number of $f$-divergences, and robust to low coverage of goals in the offline dataset.*

**Sampling from the goal-transition distribution:** Goal relabelling is an effective technique to address reward sparsity by widening the training goal distribution $q^{\texttt{train}}(g)$. It utilizes knowledge about reaching other goals, possibly unrelated to test goals, to help in reaching the test distribution of goals $q^{\texttt{test}}(g)$. In the most general case, $q^{\texttt{train}}(g)$ can be set to a uniform distribution over goals corresponding to all the states in the offline data. A common method, Hindsight Experience Replay (HER) (Andrychowicz et al., 2017) chooses a training goal distribution that depends on the current sampled state from the offline dataset as well as the data-collecting policies. In this setting, the sampling distribution used for training Eq 7, $\rho(s, a, g)$, can no longer be factorized into $\rho(s, a)$ and $q^{\texttt{train}}(g)$, as goals are conditionally dependent on state-actions. However, our formulation can naturally account for learning from such relabelled data as the SMORe objective in Eq 7 is derived considering the joint distribution $\rho(s, a, g)$. In this setting, we construct our goal transition distribution $q(s, a, g)$ as the uniform distribution over all transitions that lead to the goals selected by the HER procedure — in practice, this amounts to first selecting $g$ through HER and then selecting $\{s, a\}$ that transitions to the selected goal from the offline dataset to get a sample $\{s, a, g\}$ from goal transition distribution. We emphasize that relabelling does not change the test distribution of goals, which is an immutable property of the environment.

## 3.3 PRACTICAL ALGORITHM

To devise a stable learning algorithm we consider the Pearson $\chi^2$ divergence. Pearson $\chi^2$ divergence has been found to lead to distribution matching objectives that are stable to train as a result of a smooth quadratic generator function $f$ (Garg et al., 2021; Al-Hafez et al., 2023; Sikchi et al., 2023). Our dual formulation SMORe simplifies to the following objective:

$$
\max_{\pi_g} \min_S \overbrace{\beta(1-\gamma)\mathbb{E}_{(s,g)\sim d_0, a\sim\pi_g(\cdot|s,g)}[S(s,a,g)] + \beta\gamma\mathbb{E}_{(s,a,g)\sim q, s'\sim p(\cdot|s,a), a'\sim\pi_g(\cdot|s',g)}[S(s',a',g)]}^{\text{Decrease score at transitions under current policy } \pi_g}
$$
$$
- \underbrace{\beta\mathbb{E}_{(s,a,g)\sim q}[S(s,a,g)]}_{\text{Increase score at the proposed goal transition distribution}} + 0.25\underbrace{\mathbb{E}_{(s,a,g)\sim\text{Mix}_\beta(q,\rho)}\big[(\gamma S(s',\pi_g(s'),g) - S(s,a,g))^2\big]}_{\text{Smoothness/Bellman regularization}}. \quad (8)
$$

Equation 8 suggests a contrastive procedure, maximizing the score at the goal-transition distribution and minimizing the score at the offline data distribution under the current policy with Bellman regularization. The Bellman regularization has the interpretation of discouraging neighboring $S$ values from deviating far and smoothing the score landscape. Instantiating with KL divergence results in an objective with similar intuition while resembling an InfoNCE (Oord et al., 2018) objective. Although Propositions 1 and 2 suggest that KL divergence gives an objective that is a tighter bound to the GCRL objective, prior work has found KL divergence to be unstable in practice (Sikchi et al., 2023; Garg et al., 2023) for dual optimization.

It is important to note that $S$-function is not grounded to any rewards and does not serve as a probability density of reaching goals, but is rather a score function learned via *a Bellman-regularized contrastive learning procedure*.

We now derive a practical approach for SMORe in the offline GCRL setting. We use parameterized functions: $S_\phi(s, a, g)$, $M_\psi(s, g)$, $\pi_\theta(a|s, g)$. The offline learning regime necessitates measures to constrain the learning policy to the offline data support in order to prevent overestimation due to maximizing $\pi_g$ in Eq. 8 over potentially out-of-distribution actions. In-

---

**Algorithm 1: SMORe**

1: Init $S_\phi$, $M_\psi$, and $\pi_\theta$
2: Params: expectile $\tau$, mixture ratio $\beta$, temperature $\alpha$
3: Let $\mathcal{D} = \widehat{\rho} = \{(s, a, s', g)\}$ be an offline dataset and $q$ be goal-transition distribution
4: **for** $t = 1..T$ iterations **do**
5:     Train $S_\phi$ via Eq. 10
6:     Train $M_\psi$ via Eq. 9
7:     Update $\pi_\theta$ via Eq. 11
8: **end for**

---

spired by prior work (Kostrikov et al., 2021), we use implicit maximization to constrain the learning algorithm to learn expectiles using the observed empirical samples. More concretely, we use expectile

regression:

$$\min_{\psi} \mathcal{L}(\psi) := \mathbb{E}_{(s,a,g)\sim\rho}[L_2^{\tau}(M_{\psi}(s,g) - S_{\phi}(s,a,g))], \tag{9}$$

where $L_2^{\tau}(u) = |\tau - 1(u < 0)|u^2$. Intuitively, this step implements the maximization w.r.t $\pi$ by using expectile regression. With the above practical considerations, our objective for learning $S_{\phi}$ reduces to:

$$\min_{\phi} \mathcal{L}(\phi) := \beta(1-\gamma)\mathbb{E}_{(s,g)\sim\mathcal{D},a\sim\pi_g(\cdot|a,g)}[S_{\phi}(s,\pi_g(s),g)] + \beta\gamma\mathbb{E}_{(s,a,g)\sim q,s'\sim p(\cdot|s,a)}[S_{\phi}(s',\pi_g(s'),g)]$$

$$- \beta\mathbb{E}_{(s,a,g)\sim q}[S_{\phi}(s,a,g)] + \mathbb{E}_{(s,a,g)\sim\mathtt{Mix}_{\beta}(q,\rho)}[(\gamma M_{\psi}(s',g) - S_{\phi}(s,a,g))^2], \tag{10}$$

where we have set the offline data distribution as our initial state distribution. Finally, the policy is extracted via advantage-weighted regression that learns in-distribution actions maximizing the score $S(s,a,g)$:

$$\min_{\theta} \mathcal{L}(\theta) := \mathbb{E}_{(s,a,g)\sim\rho}[\exp(\alpha(S_{\phi}(s,a,g) - M_{\psi}(s,g)))\log(\pi_{\theta}(a|s,g))], \tag{11}$$

where $\alpha$ is the temperature parameter. Algorithm 1 details the practical implementation.

# 4 EXPERIMENTS

Our experiments study the effectiveness of proposed GCRL algorithm `SMORe` on a set of simulated benchmarks against other GCRL methods that employ behavior cloning, RL with sparse reward, and contrastive learning. We also analyze if `SMORe` is robust to environment stochasticity — a number of prior methods are based on an assumption of deterministic dynamics. Then, we study if the discriminator-free nature of `SMORe` is indeed able to prevent performance degradation in the face of low expert coverage in offline data. Finally, we analyze if `SMORe`'s score-modeling approach helps `SMORe` scale to a vision-based manipulation offline GCRL benchmark, as density modeling and discriminator learning become increasingly difficult with high-dimensional observations. Hyperparameter ablations can be found in Appendix D.

## 4.1 EXPERIMENTAL SETUP

Our experiments will use a suite of simulated goal-conditioned tasks extending the tasks from previous work (Ma et al., 2022; Plappert et al., 2018). In particular we consider the following environments: `Reacher`, Robotic arm environments - [`SawyerReach`, `SawyerDoor`, `FetchReach`, `FetchPick`, `FetchPush`, `FetchSlide`], Anthropomorphic hand environment - `HandReach` and Locomotion environments - [`CheetahTgtVel-me`, `CheetahTgtVel-re`, `AntTgtVel-me`, `AntTgtVel-re`]. Tasks in all environments are specified by a sparse reward function. Depending on whether the task involves object manipulation, the goal distribution is defined over valid configurations in robot or object space. The offline dataset for manipulation tasks consists of transitions collected by a random policy or mixture of 90% random policy and 10% expert policy. For locomotion tasks, we generate our dataset using the D4RL benchmark (Fu et al., 2020), combining a random or medium dataset with 30 episodes of expert data. Note that the policies used to collect the expert locomotion datasets have a different objective than the tasks here, which are to achieve and maintain a particular desired velocity.

## 4.2 OFFLINE GOAL-CONDITIONED RL BENCHMARK

**Baselines.** We compare to state-of-art offline GCRL algorithms, consisting of both regression-based and actor-critic methods. The occupancy-matching based methods are: (1) **GoFar** (Ma et al., 2022), which derives a dual objective for GCRL based on a coverage assumption. The behavior cloning based methods are: (1) **GCSL** (Ghosh et al., 2019), which incorporates hindsight relabeling in conjunction with behavior cloning to clone actions that lead to a specified goal, and (2) **WGCSL** (Yang et al., 2022), which improves upon GCSL by incorporating discount factor and advantage weighting into the supervised policy learning update. **Contrastive RL** (Eysenbach et al., 2022b) generalizes C-learning (Eysenbach et al., 2020) and represents contrastive GCRL approaches. The RL with sparse reward methods are (1) **IQL** (Kostrikov et al., 2021) where we use a state-of-the-art offline RL method repurposed for GCRL along with HER (Andrychowicz et al., 2017) goal sampling, and (2)

| Task | Occupancy Matching | | Behavior cloning | | Contrastive RL | RL+sparse reward | |
|------|------|------|------|------|------|------|------|
| | SMORe | GoFAR | WGCSL | GCSL | CRL | AM | IQL |
| Reacher (⋆) | **28.40**±0.88 | 19.74±1.35 | 17.57±0.53 | 15.87±1.31 | 16.44±0.60 | 23.26 ±0.14 | 11.70 ±1.97 |
| SawyerReach (⋆) | **37.67**±0.12 | 15.34±0.64 | 15.15±0.44 | 14.25±0.7 | 22.32 ±0.34 | 23.34±0.17 | 35.18 ±0.29 |
| SawyerDoor (⋆) | **31.48**±0.46 | 18.94±0.01 | 20.01±1.55 | 20.88±0.22 | 12.96±5.19 | 22.12 ±0.13 | 25.52 ±1.45 |
| FetchReach (⋆) | **35.08**± 0.54 | 28.2 ±0.61 | 21.9± 2.13 | 20.91 ± 2.78 | 30.07±0.07 | 30.1 ± 0.32 | 34.43 ± 1.00 |
| FetchPick (⋆) | **26.47** ± 1.34 | 19.7 ± 2.57 | 9.84 ± 2.58 | 7.58±1.85 | 0.42±0.29 | 8.94 ± 3.09 | 16.8 ± 3.10 |
| FetchPush (⋆) | **26.83**± 1.21 | 18.2 ± 3.00 | 14.7 ± 2.65 | 13.4 ± 3.02 | 2.40 ±1.28 | 14.0 ± 2.81 | 22.40 ± 0.74 |
| FetchSlide | **4.99**± 0.40 | 2.47 ± 1.44 | 2.73 ± 1.64 | 1.75 ± 1.3 | 0.0±0.0 | 1.46 ± 1.38 | **4.80** ± 1.59 |
| HandReach (⋆) | **18.68** ± 3.35 | 11.5 ± 5.26 | 5.97 ± 4.81 | 1.37 ± 2.21 | 0.0±0.0 | 0.0 ± 0.0 | 1.44 ± 1.77 |
| CheetahTgtVel-m-e (⋆) | **136.71** ± 10.59 | 0.0± 0.0 | 0.0± 0.0 | 95.98± 15.72 | 0.0±0.0 | 0.0± 0.0 | 100.38± 1.22 |
| CheetahTgtVel-r-e (⋆) | **60.01** ± 39.40 | 0.0± 0.0 | 0.0± 0.0 | 11.56 ± 13.47 | 0.0±0.0 | 0.0± 0.0 | 0.0± 0.0 |
| AntTgtVel-m-e | 154.95± 19.44 | **168.27**± 9.58 | 0.0± 0.0 | 164.54± 7.69 | 0.0±0.0 | 0.0± 0.0 | 148.17 ± 5.43 |
| AntTgtVel-r-e (⋆) | **126.22**± 14.40 | 74.36± 15.97 | 0.0± 0.0 | 104.95± 6.00 | 0.0±0.0 | 0.0± 0.0 | 3.06 ± 2.64 |

Table 1: Discounted Return for the offline GCRL benchmark. Results are averaged over 10 seeds. 'm-e' and 'r-e' stands for medium-expert mixture and random-expert mixture respectively. (⋆) denotes statistically significant improvements.

**ActionableModel (AM)** (Chebotar et al., 2021), which incorporates conservative Q-Learning (Kumar et al., 2020) as well as goal-chaining on top of an actor-critic method.

The results for all baselines are tuned individually, particularly the best HER ratio was searched among $\{0.2, 0.5, 0.8, 1.0\}$ for each task. SMORe shares the same network architecture for baselines and uses a mixture ratio of $\beta = 0.5$. Each method is trained for 10 seeds. Complete architecture and hyperparameter table as well as additional training details are provided in Appendix C.

Table 1 reports the **discounted return** obtained by the learned policy with a sparse binary task reward. (⋆) denotes statistically significant improvement over the second best method under a Mann-Whitney U test with a significance level of 0.05. This metric allows us to compare the algorithms on a finer scale to understand which methods reach the goal as fast as possible and stay in the goal region thereafter for the longest time. Additional results on metrics like success rate and final distance to goal can be found in the appendix. These additional metrics do not take into consideration how *precisely* and *consistently* a goal is being reached. In Table 1, we see that SMORe enjoys a high-performance gain consistently across all tasks in the extended offline GCRL benchmark.

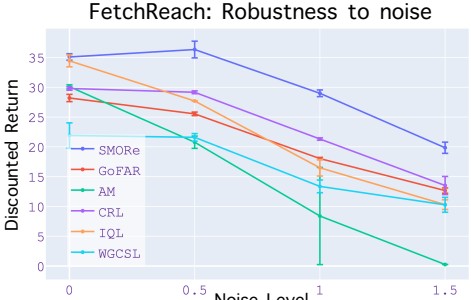

Figure 2: SMORe is robust in stochastic environments. With increasing noise, SMORe still outperforms prior methods.

**Robustness to environment stochasticity:** We consider a noisy version of the FetchReach environment in this experiment. Gaussian zero-mean noise is added before executing an action to generate different variants of the environment with standard deviations of $\{0.5, 1.0, 1.5\}$. Datasets for these environments are obtained from prior work (Ma et al., 2022). As we see in Figure 2, SMORe is robust to stochasticity in the environment, outperforming baselines in terms of discounted return. Behavior cloning based approaches assume deterministic dynamics and are therefore over-optimistic in stochastic environments.

### 4.3 ROBUSTNESS OF OCCUPANCY-MATCHING METHODS TO DECREASING EXPERT COVERAGE

We posit that the discriminator-free nature of SMORe makes it more robust to decreasing goal coverage, as it does not suffer from cascading errors stemming from a learned discriminator. In this section, we set out to test this hypothesis by decreasing the amount of expert data in the offline goal-reaching dataset. We compare with GoFAR in Table 2 due to the similarity between methods and GoFAR's restrictive assumption on coverage of expert data in the suboptimal dataset. Comparison against all the baselines can be found in Appendix D.

Our hypothesis holds true as we see in Table 2, the performance of the discriminator-based method GoFar rapidly decays as expert data is decreased in the offline dataset – 28.4% with 2.5% and 36.15% with 1% expert data(i.e. optimal policy's coverage) respectively. SMORe shows a much slower decay

| Task | 5 % expert data | | 2.5 % expert data | | 1 % expert data | |
|------|------|------|------|------|------|------|
| | SMORe | GoFAR | SMORe | GoFAR | SMORe | GoFAR |
| Reacher | $22.43_{\pm 3.46}$ | $16.86_{\pm 1.26}$ | $17.92 \pm _{0.93}$ | $12.20_{\pm 0.81}$ | $19.61 \pm _{1.56}$ | $11.52 \pm _{0.52}$ |
| SawyerReach | $36.35_{\pm 0.37}$ | $13.20_{\pm 1.36}$ | $36.74_{\pm 0.62}$ | $11.57_{\pm 1.79}$ | $35.44_{0.27}$ | $9.34_{\pm 0.17}$ |
| SawyerDoor | $32.82_{\pm 0.88}$ | $20.07_{\pm 0.01}$ | $25.69_{\pm 0.21}$ | $19.54_{\pm 1.32}$ | $23.78_{\pm 2.88}$ | $18.04_{\pm 1.80}$ |
| FetchReach | $36.00_{\pm 0.01}$ | $27.66 \pm _{0.55}$ | $35.58 \pm _{0.47}$ | $27.84 \pm _{0.82}$ | $35.97 \pm _{0.25}$ | $28.01 \pm _{0.20}$ |
| FetchPick | $26.43_{\pm 1.95}$ | $16.21 \pm _{1.46}$ | $26.17_{\pm 3.37}$ | $3.21 \pm _{2.22}$ | $15.38 \pm _{1.52}$ | $0.31 \pm _{0.31}$ |
| FetchPush | $23.81_{\pm 0.37}$ | $18.2 \pm _{3.00}$ | $22.75_{\pm 1.08}$ | $5.17 \pm _{2.01}$ | $19.04_{\pm 2.79}$ | $4.23_{\pm 3.96}$ |
| FetchSlide | $4.05_{\pm 1.12}$ | $1.08 \pm _{0.06}$ | $3.11 \pm _{1.61}$ | $0.96 \pm _{0.73}$ | $3.50_{\pm 0.97}$ | $0.86 \pm _{1.22}$ |
| Average Performance | 25.98 | 16.18 | 23.99 | 11.49 | 21.81 | 10.33 |
| Avg. Perf. Drop | 0 | 0 | -7.6% | -28.4% | -16% | -36.15% |

Table 2: Discounted Return for the offline GCRL benchmark with 5%, 2.5% and 1% expert data in offline dataset. Results are averaged over 10 seeds.

in performance, 7.6% with 2.5% and 16% with 1% expert data, attesting to the method's robustness under decreasing expert coverage in the offline dataset.

## 4.4 OFFLINE GCRL WITH IMAGE OBSERVATIONS

SMORe provides an effective algorithm for offline GCRL in high-dimensional observation spaces by learning unnormalized scores using a contrastive procedure as opposed to prior works that learn normalized densities (Eysenbach et al., 2020) which are difficult to learn or density ratios (Eysenbach et al., 2022b; Zheng et al., 2023) which do not optimize for the optimal goal-conditioned policy in the offline GCRL setting. Similar to prior work (Eysenbach et al., 2022b), we consider the following structure in S-function parameterization to learn performant and generalizable policies: $S(s, a, g) = \phi(s, a)^T \psi(g)$. The S-function can be interpreted as the similarity between the two representations given by $\phi$ and $\psi$. Our network architecture for both representations is similar to Zheng et al. (2023) and is kept the same across all baselines to ensure a fair comparison of the underlying GCRL method.

We use the offline GCRL benchmark from (Zheng et al., 2021) which learns goal-reaching policies from an image-observation dataset of 250K transitions with the horizon ranging from 50-100. The benchmark adds another layer of complexity by testing on goals absent from the dataset — the dataset contains primitive behaviors like picking up objects and pushing drawers but no behavior that completes the compound task we consider from the initial state. The observations and goals are 48x48x3 RGB images.

**Baselines** We compare to the best performing GCRL algorithms from Section 1 as well as a recent state-of-the-art work, stable contrastive RL Zheng et al. (2023). Stable contrastive RL features a number of improvements over contrastive RL by changing design decisions in neural network architecture, layer normalization, and data augmentation. Since our objective is to compare the quality of the underlying GCRL algorithm, we keep these design decisions consistent across the board.

**Results** Figure 3 shows the success rate on a variety of unseen tasks for all the methods. SMORe achieves highest success rates across all the tasks, even for the most challenging task of pick, place

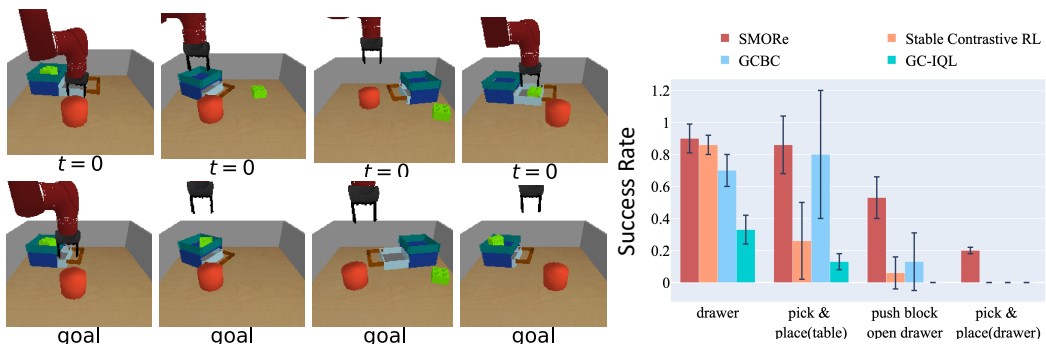

Figure 3: Evaluation on simulated manipulation tasks with image observations. The left image shows the starting state at the top and the goal at the bottom for evaluation tasks. The error bars show the standard deviation with 5 random seeds. SMORe is competitive or outperforms prior methods on all the tasks we considered.

and closing the drawer. We note that our results differ from Zheng et al. (2023) for the baselines as we apply the same design decisions for all methods whereas Zheng et al. (2023) focuses on ablating design decisions.

## 5 RELATED WORKS

**Offline Goal Conditioned Reinforcement Learning.** Learning to achieve goals in the environment optimally forms the basis of goal-condition RL problems. Studies in cognitive science (Molinaro and Collins, 2023) underscore the importance goal-achieving plays in human development. Offline GCRL approaches are typically catered to designing learning algorithms for addressing the sparsity of reward function in the offline setting. One of the most successful techniques in this setting has been hindsight relabelling. Hindsight-experience relabelling (HER) (Kaelbling, 1993; Andrychowicz et al., 2017) suggests relabelling any experience with some commanded goal to the goal that was actually achieved in order to leverage generalization. HER has been investigated in the setting of learning from demonstrations (Ding et al., 2019) and exploration (Fang et al., 2019) to validate its effectiveness. A number of prior works (Ghosh et al., 2019; Yang et al., 2019; Chen et al., 2020; Ding et al., 2019; Lynch et al., 2020; Paster et al., 2020; Srivastava et al., 2019; Hejna et al., 2023) have investigated using goal-conditioned behavior cloning, a strategy that uses relabelling to learn goal-conditioned policies, as a way to learn performant policies. Eysenbach et al. (2022a) shows that this line of work has a limitation of learning suboptimal policies that do not consistently improve over the policy that collected the dataset. The simplest strategy of applying single-task RL to the problem of multi-task goal reaching requires learning a $Q$-function which represents normalized densities over the state-action space. Contrastive RL (Eysenbach et al., 2022b; 2020; Zheng et al., 2023) emerged as another alternative for GCRL which relabels trajectories and, rather than use that relabelling to learn policies, learns a $Q$-function using a contrastive procedure. While these approaches learn optimal policies in the online setting, they fall behind in the offline setting where they only learn a policy that greedily improves over the $Q$-function of the data collecting policy. Our work learns optimal policies by presenting an off-policy objective that solves GCRL and furthermore learns scores (or unnormalized densities) that alleviate the learning challenges of normalized density estimation.

**Distribution matching.** Our approach is inspired by the distribution matching approach (Ghasemipour et al., 2020; Ni et al., 2021; Sikchi et al., 2022; Swamy et al., 2021; Sikchi et al., 2023) prominent in imitation learning. Ghasemipour et al. (2020); Ni et al. (2021) takes the problem of imitating an expert demonstrator in the environment and converts it into a problem of distribution matching between the current policy's state-action visitation distribution and the expert policy's visitation distribution. Indeed, a prior work $f$-PG (Agarwal et al., 2024) proposes a distribution matching approach to GCRL but is restricted to the on-policy setting. Another, prior work (Ma et al., 2022) creates one such distribution matching problem and presents a new optimization problem for GCRL in the form of an off-policy dual (Nachum and Dai, 2020; Sikchi et al., 2023). Such an off-policy dual is very appealing for the offline RL setup, as optimizing for this dual only requires sampling from the offline data distribution. A limitation of their dual construction is the fact that they require learning a discriminator and use that discriminator as the pseudo-reward for solving the GCRL objective. Our approach presents a new construction for GCRL as a distribution matching problem along with a dual construction that leads to a more performant discriminator-free off-policy approach for GCRL.

## 6 CONCLUSION

Prior work in performant online goal-conditioned RL often relies on iterated behavior cloning or contrastive RL. However, these approaches are suboptimal for the offline setting. Existing methods specifically derived for offline GCRL require learning a discriminator and using it as a pseudo-reward, enabling compounding errors that make the resulting policy ineffective. We present an occupancy-matching approach to offline GCRL that provably optimizes a lower bound to the regularized GCRL objective. Our method is discriminator-free, applicable to a number of $f$-divergences, and learns unnormalized scores over actions at a state to reach the goal. We show that these positive aspects of our algorithm allow us to empirically outperform prior methods, stay robust under decreasing goal coverage, and scale to high-dimensional observation space for GCRL.

ACKNOWLEDGEMENTS

We thank the MIDI lab members and ICLR reviewers for valuable feedback on this work. This work has taken place in the Safe, Correct, and Aligned Learning and Robotics Lab (SCALAR) at The University of Massachusetts Amherst and Machine Intelligence through Decision-making and Interaction (MIDI) Lab at The University of Texas at Austin. SCALAR research is supported in part by the NSF (IIS-2323384), AFOSR (FA9550-20-1-0077), and ARO (78372-CS, W911NF-19-2-0333), and the Center for AI Safety (CAIS). This research was also sponsored by the Army Research Office under Cooperative Agreement Number W911NF-19-2-0333. HS and AZ are funded in part by a sponsored research agreement with Cisco Systems Inc. The views and conclusions contained in this document are those of the authors and should not be interpreted as representing the official policies, either expressed or implied, of the Army Research Office or the U.S. Government. The U.S. Government is authorized to reproduce and distribute reprints for Government purposes notwithstanding any copyright notation herein.

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

# A APPENDIX

## A.1 THEORY

In this section, we first show the equivalence of the GCRL problem and the distribution-matching objective of imitation learning. Then, we show how the mixture distribution objective relates to offline GCRL objective. Finally, we derive the dual objective for mixture distribution matching that leads to our method SMORe.

### A.1.1 REDUCTION OF GCRL TO DISTRIBUTION MATCHING

**Proposition 1.** *Consider a stochastic MDP, a stochastic policy $\pi$, and a sparse reward function $r(s, a, g) = \mathbb{E}_{s' \sim p(\cdot|s,a)}\left[\mathbb{I}(\phi(s') = g, q^{train}(g) > 0)\right]$ where $\mathbb{I}$ is an indicator function. Define a soft goal transition distribution to be $q(s, a, g) \propto exp(\alpha\ r(s, a, g))$. The following bounds hold for any $f$-divergence that upper bounds KL-divergence (eg. $\chi^2$, Jensen-Shannon):*

$$J^{train}(\pi_g) + \frac{1}{\alpha}\mathcal{H}(d^{\pi_g}) \geqslant -\frac{1}{\alpha}\mathcal{D}_f(d^{\pi_g}(s, a, g)\|q(s, a, g)) + C, \tag{4}$$

*where $\mathcal{H}$ denotes the entropy, $\alpha$ is a temperature parameter and $C$ is the partition function for $e^{R(s,a,g)}$. Furthermore, the bound is tight when $f$ is the KL-divergence.*

*Proof.* This proof is adapted from Ma et al. (2022) for goal transition distributions and state-action distributions. Let $Z = \int e^{R(s,a,g)}\ ds\ da\ dg$ and $\alpha > 0$ be the temperatue parameter. Note that $q(s, a, g) = e^{r(s,a,g)}$ where r is defined in the proposition, strictly generalizes the original definition $q(s, a, g) = q^{train}(g)\mathbb{E}_{s' \sim p(\cdot|s,a)}[\mathbb{I}(\phi(s') = g)]$ and recovers it when $\alpha \to \infty$. Starting with the true GCRL objective:

$$\alpha J(\pi_g) = \mathbb{E}_{d^{\pi_g}}\left[\alpha R(s, a, g)\right] \tag{12}$$

$$= \mathbb{E}_{d^{\pi_g}}\left[\log e^{\alpha R(s,a,g)}\right] \tag{13}$$

$$= \mathbb{E}_{d^{\pi_g}}\left[\log\left(\frac{e^{\alpha R(s,a,g)}}{Z}\frac{d^{\pi_g}(s, a, g)}{d^{\pi_g}(s, a, g)}Z\right)\right] \tag{14}$$

$$= \mathbb{E}_{d^{\pi_g}}\left[\log\left(\frac{q(s, a, g)}{d^{\pi_g}(s, a, g)}Z\right)\right] + \mathbb{E}_{d^{\pi_g}}\left[\log d^{\pi_g}\right] \tag{15}$$

$$= -D_{KL}(d^{\pi_g}(s, a, g)\|q(s, a, g)) - \mathcal{H}(d^{\pi_g}) + \log(Z) \tag{16}$$

Rearranging terms we get:

$$J(\pi_g) + \frac{1}{\alpha}\mathcal{H}(d^{\pi_g}) = -\frac{1}{\alpha}D_{KL}(d^{\pi_g}(s, a, g)\|q(s, a, g)) + C \tag{17}$$

For any $f$-divergence that upper bounds the KL divergence we have:

$$J(\pi_g) + \frac{1}{\alpha}\mathcal{H}(d^{\pi_g}) = -\frac{1}{\alpha}D_{KL}(d^{\pi_g}(s, a, g)\|q(s, a, g)) + C \geqslant -\frac{1}{\alpha}D_f(d^{\pi_g}(s, a, g)\|q(s, a, g)) + C \tag{18}$$

$\square$

**A (dataset) regularized GCRL objective:** Define a regularized objective for GCRL as follows:

$$J_{offline}(\pi) = \alpha_1\mathbb{E}_{d^\pi}\left[e^{r(s,a,g)}\right] + \alpha_2\mathbb{E}_{d^\pi(s,a,g)}[\rho(s, a, g)]. \tag{19}$$

The second term in the objective $\mathbb{E}_{d^\pi(s,a,g)}[\rho(s, a, g)]$ above is maximized when the policy visitation places more probability mass on the most visited transitions in the dataset. To see why this is, consider two probability distributions represented as vectors $d^\pi$ and $\rho$ with individuals elements of the vector indexed by $i$:

$$\langle d^\pi, \rho \rangle \leqslant \max_i \rho_i \tag{20}$$

The equality holds only when $d^\pi$ places probability mass on all state-action-goal tuples which are most visited in the offline dataset $\rho$. The first term maximizes the true GCRL objective while the second term prefers staying close to transitions that are most frequently observed in the offline dataset. A constraint of $\langle d^\pi, \rho \rangle \geqslant 1 - \delta$ implies that the agent visitation places atleast half of the probability mass on state-action-goal tuples whose average visitation in offline dataset is greater than or equal

to $1 - \delta$. With weights $\alpha_1$ and $\alpha_2$, the objective above reflects an lagrangian relaxation to this constraint. Thus the above offline objective presents an alternative offline objective when compared to the classical offline RL objectives Wu et al. (2019); Nachum and Dai (2020).

Proposition 2 derives the connection between the dataset regularized GCRL objective and `SMORe`:

**Proposition 2.** *Consider a stochastic MDP, a stochastic policy $\pi$, and a sparse reward function $r(s, a, g) = \mathbb{E}_{s' \sim p(\cdot|s,a)}\big[\mathbb{I}(\phi(s') = g, q^{train}(g) > 0)\big]$ where $\mathbb{I}$ is an indicator function, define a soft goal transition distribution to be $q(s, a, g) \propto exp(\alpha\ r(s, a, g))$ the following bounds hold for any $f$-divergence that upper bounds KL-divergence (eg. $\chi^2$, Jensen-Shannon):*

$$\log J_{offline}(\pi_g) + \mathcal{H}(\texttt{Mix}_\beta(d, \rho)(s, a, g)) + C \geqslant -\mathcal{D}_f(\texttt{Mix}_\beta(d, \rho)(s, a, g) \| \texttt{Mix}_\beta(q, \rho)(s, a, g)),\tag{21}$$

*where $\mathcal{H}$ denotes the entropy, $\alpha$ is a temperature parameter, $\alpha_1 = \beta^2$, $\alpha_2 = \beta(1 - \beta)Z$ and $C$ is a positive constant. Furthermore, the bound is tight when $f$ is the KL-divergence.*

*Proof.* We first consider the following two objectives for GCRL and show that they are equivalent. This reduction will later help in proving a connection to mixture occupancy matching. We consider $\alpha = 1$ w.l.o.g. Here are two objectives we consider:

$$J(\pi) = \mathbb{E}_{d^\pi}[r(s, a, g)]\tag{22}$$

$$J'(\pi) = \mathbb{E}_{d^\pi}\Big[e^{r(s,a,g)}\Big]\tag{23}$$

In GCRL reward functions are sparse and binary. We show the equivalence of first two objectives in find the optimal goal conditioned policy via two arguments. First, notice that the rewards for goal transition states for objective $J'(\pi)$ is $e$ and 1 for all other transitions. This is in contrast to $J(\pi)$ which considers a reward function 1 at goal transitions states and 0 otherwise. Under our assumption of infinite horizon discounted MDP, we can translate the rewards while keeping the optimal policy same in MDP considered by $J'(\pi)$ to $e - 1$ at goal transitions states and 0 otherwise. Further we can scale the rewards by $1/(e - 1)$ and recover and MDP with same optimal policy that has reward of 1 at goal-transition states and 0 otherwise. This concludes the equivalence of maximizing $J'(\pi)$ as an alternative to $J(\pi)$ while recovering the same optimal policy.

We now consider a regularized (pessimistic/offline) GCRL problem with the shifted reward functions $e^{r(s,a,g)}$ that maximizes the reward while ensuring the policy visitation stays close to offline data visitation in cosine similarity.

$$J_{offline}(\pi) = \alpha_1 \mathbb{E}_{d^\pi}\Big[e^{r(s,a,g)}\Big] + \alpha_2 \mathbb{E}_{d^\pi(s,a,g)}[\rho(s, a, g)].\tag{24}$$

With a particular instantiation of hyperparameters we show that the $J_{offline}(\pi)$ objective can be simplified to an equivalent objective $J'_{offline}(\pi)$ by setting $\alpha_1 = \beta^2$ and $\alpha_2 = \beta(1 - \beta)Z$ where $Z$ is the partition function for $e^{r(s,a,g)}$ over entire $\mathcal{S} \times \mathcal{A} \times \mathcal{G}$.

$$J'_{offline}(\pi) = \mathbb{E}_{\texttt{Mix}_\beta(d,\rho)(s,a,g)}\Big[\beta e^{r(s,a,g)} + (1 - \beta)\rho(s, a, g).Z\Big]\tag{25}$$

$$J'_{offline}(\pi) = \mathbb{E}_{\texttt{Mix}_\beta(d,\rho)(s,a,g)}\Big[\beta e^{r(s,a,g)} + (1 - \beta)\rho(s, a, g).Z\Big]\tag{26}$$

$$= \beta^2 \mathbb{E}_{d^\pi}\Big[e^{r(s,a,g)}\Big] + \beta(1 - \beta)Z\mathbb{E}_{d^\pi}[\rho(s, a, g)]\tag{27}$$

$$+ (1 - \beta)\mathbb{E}_{d^o}\Big[\beta e^{r(s,a,g)} + (1 - \beta)\rho(s, a, g).Z\Big]\beta\tag{28}$$

$$\tag{29}$$

$$= \beta^2 \mathbb{E}_{d^\pi}\Big[e^{r(s,a,g)}\Big] + \beta(1 - \beta)Z\mathbb{E}_{d^\pi}[\rho(s, a, g)] + C'\tag{30}$$

$$= J_{offline}(\pi) + C'\tag{31}$$

Now that we have shown $J'_{offline}(\pi) \equiv J_{offline}(\pi)$ and hence solving the same optimization problem, we proceed to derive connections with mixture occupancy matching which follows through an application of Jensen's inequality:

$$\log J'_{offline}(\pi) = \log \mathbb{E}_{\text{Mix}_\beta(d,\rho)(s,a,g)}\Big[\beta e^{r(s,a,g)} + (1-\beta)\rho(s,a,g).Z\Big] \tag{32}$$

$$\geqslant \mathbb{E}_{\text{Mix}_\beta(d,\rho)(s,a,g)}\Big[\log(\beta e^{r(s,a,g)} + (1-\beta)\rho(s,a,g).Z)\Big] \tag{33}$$

$$\tag{34}$$

$$= \mathbb{E}_{\text{Mix}_\beta(d,\rho)(s,a,g)}[\log(\beta q(s,a,g) + (1-\beta)\rho(s,a,g))] + \log Z \tag{35}$$

$$= -D_{KL}[\text{Mix}_\beta(d,\rho)(s,a,g)\|\text{Mix}_\beta(q,\rho)(s,a,g)] - \mathcal{H}(\text{Mix}_\beta(d,\rho)(s,a,g)) + \log Z \tag{36}$$

For any $f$-divergence that upperbounds the KL divergence since $Z \geqslant 1$ we have:

$$\log J'_{offline}(\pi) + \frac{1}{\alpha}\mathcal{H}(\text{Mix}_\beta(d,\rho)(s,a,g)) \geqslant -\frac{1}{\alpha}D_f(\text{Mix}_\beta(d,\rho)(s,a,g)\|\text{Mix}_\beta(q,\rho)(s,a,g)) \tag{37}$$

Further simplifying using Eq 31:

$$\log J_{offline}(\pi) + \frac{1}{\alpha}\mathcal{H}(\text{Mix}_\beta(d,\rho)(s,a,g)) + C \geqslant -\frac{1}{\alpha}D_f(\text{Mix}_\beta(d,\rho)(s,a,g)\|\text{Mix}_\beta(q,\rho)(s,a,g)) \tag{38}$$

$$\square$$

Optimizing the mixture distribution matching objective of SMORe maximizes a variant of *of-fline/dataset regularized* GCRL objective where the entropy for distribution $\text{Mix}_\beta(d,\rho)(s,a,g)$ is jointly maximized. Therefore we have shown that the minimizing discrepancy of mixture distribution occupancy maximizes a lower bounds to an offline variant of maxent GCRL objective.

## A.2 Convex Conjugates and $f$-divergences

We first review the basics of duality in reinforcement learning. Let $f : \mathbb{R}_+ \to \mathbb{R}$ be a convex function. The convex conjugate $f^* : \mathbb{R}_+ \to \mathbb{R}$ of $f$ is defined by:

$$f^*(y) = \sup_{x\in\mathbb{R}_+}[xy - f(x)]. \tag{39}$$

The convex conjugates have the important property that $f^*$ is also convex and the convex conjugate of $f^*$ retrieves back the original function $f$. We also note an important relation regarding $f$ and $f^*$: $(f^*)' = (f')^{-1}$, where the $'$ notation denotes first derivative.

Going forward, we would be dealing extensively with $f$-divergences. Informally, $f$-divergences (Rényi, 1961) are a measure of distance between two probability distributions. Here's a more formal definition:

Let $P$ and $Q$ be two probability distributions over a space $\mathcal{Z}$ such that $P$ is absolutely continuous with respect to $Q$ [4]. For a function $f : \mathbb{R}_+ \to \mathbb{R}$ that is a convex lower semi-continuous and $f(1) = 0$, the $f$-divergence of $P$ from $Q$ is

$$D_f(P \,\|\, Q) = \mathbb{E}_{z\sim Q}\left[f\left(\frac{P(z)}{Q(z)}\right)\right]. \tag{40}$$

Table 3 lists some common $f$-divergences with their generator functions $f$ and the conjugate functions $f^*$.

## A.3 SMORe: Dual objective for Offline Goal conditioned reinforcement learning

In this section, we derive the dual objective for solving the multi-task occupancy problem formulation for GCRL. First, we derive the original variant of SMORe for the GCRL problem and later derive the action-free SMORe variant for the interested readers.

---

[4]Let $z$ denote the random variable. For any measurable set $Z \subseteq \mathcal{Z}$, $Q(z \in Z) = 0$ implies $P(z \in Z) = 0$.

| Divergence Name | Generator $f(x)$ | Conjugate $f^*(y)$ |
|---|---|---|
| KL (Reverse) | $x \log x$ | $e^{(y-1)}$ |
| Squared Hellinger | $(\sqrt{x} - 1)^2$ | $\frac{y}{1-y}$ |
| Pearson $\chi^2$ | $(x-1)^2$ | $y + \frac{y^2}{4}$ |
| Total Variation | $\frac{1}{2}|x-1|$ | $y$ if $y \in \left[-\frac{1}{2}, \frac{1}{2}\right]$ otherwise $\infty$ |
| Jensen-Shannon | $-(x+1)\log(\frac{x+1}{2}) + x \log x$ | $-\log(2 - e^y)$ |

Table 3: List of common $f$-divergences.

**Theorem 1.** *The dual problem to the primal occupancy matching objective (Equation 6) is given by:*

$$\max_{\pi_g} \min_{S} \beta(1-\gamma)\mathbb{E}_{d_0,\pi_g}[S(s,a,g)] + \mathbb{E}_{\text{Mix}_\beta(q,\rho)}[f^*(\gamma P^{\pi_g}S(s,a,g) - S(s,a,g))] \quad (7)$$

$$- (1-\beta)\mathbb{E}_\rho[\gamma P^{\pi_g}S(s,a,g) - S(s,a,g)],$$

*where $f^*$ is conjugate function of $f$ and $S$ is the Lagrange dual variable defined as $S : \mathcal{S} \times \mathcal{A} \times \mathcal{G} \to \mathbb{R}$. Moreover, as strong duality holds from Slater's conditions the primal and dual share the same optimal solution $\pi_g^*$ for any offline transition distribution $\rho$.*

*Proof.* Recall that: $\text{Mix}_\beta(d,\rho)(s,a,g) := \beta d(s,a,g) + (1-\beta)\rho(s,a,g)$ and $\text{Mix}_\beta(q,\rho)(s,a,g) := \beta q(s,a,g) + (1-\beta)\rho(s,a,g)$. $\text{Mix}_\beta(d,\rho)(s,a,g)$ denotes the mixture between the current agent's joint-goal visitation distribution with an offline transition dataset potentially suboptimal and $\text{Mix}_\beta(q,\rho)(s,a,g)$ is the mixture between the expert's visitation distribution with arbitrary experience from the offline transition dataset. Minimizing the divergence between these visitation distributions still solves the occupancy problem, i.e $d^{\pi_g} = q$ when $q$ is achievable. We start with the primal formulation from Eq 6 for mixture divergence regularization:

$$\max_{d(s,a,g) \geqslant 0, \pi(a|s)} -D_f(\text{Mix}_\beta(d,\rho)(s,a,g) \,||\, \text{Mix}_\beta(q,\rho)(s,a,g))$$

$$\text{s.t } d(s,a,g) = (1-\gamma)\rho_0(s,g).\pi(a|s,g) + \gamma\pi(a|s,g)\sum_{s',a'} d(s',a',g)p(s|s',a').$$

Applying Lagrangian duality and convex conjugate (39) to this problem, we can convert it to an unconstrained problem with dual variables $S(s,a,g)$ defined for all $s,a \in \mathcal{S} \times \mathcal{A} \times \mathcal{G}$:

$$\max_{\pi,d \geqslant 0} \min_{S(s,a,g)} -D_f(\text{Mix}_\beta(d,\rho)(s,a,g) \,||\, \text{Mix}_\beta(q,\rho)(s,a,g))$$

$$+ \sum_{s,a,g} S(s,a,g)\left((1-\gamma)d_0(s,g).\pi(a|s,g) + \gamma\sum_{s',a'} d(s',a',g)p(s|s',a')\pi(a|s,g) - d(s,a,g)\right) \tag{41}$$

$$= \max_{\pi,d \geqslant 0} \min_{S(s,a,g)} (1-\gamma)\mathbb{E}_{d_0(s,g),\pi(a|s,g)}[S(s,a,g)]$$

$$+ \mathbb{E}_{s,a,g \sim d}\left[\gamma\sum_{s',a'} p(s'|s,a)\pi(a'|s')S(s',a',g) - S(s,a,g)\right] \tag{42}$$

$$- D_f(\text{Mix}_\beta(d,\rho)(s,a,g) \,||\, \text{Mix}_\beta(q,\rho)(s,a,g)) \tag{43}$$

$$= \max_{\pi,d \geqslant 0} \min_{S(s,a,g)} \beta(1-\gamma)\mathbb{E}_{d_0(s,g),\pi(a|s,g)}[S(s,a,g)]$$

$$+ \beta\mathbb{E}_{s,a,g \sim d}\left[\gamma\sum_{s',a'} p(s'|s,a)\pi(a'|s')S(s',a',g) - S(s,a,g)\right] \tag{44}$$

$$+ (1-\beta)\mathbb{E}_{s,a,g\sim\rho}\left[\gamma\sum_{s',a'}p(s'|s,a)\pi(a'|s')S(s',a',g) - S(s,a,g)\right]$$

$$- (1-\beta)\mathbb{E}_{s,a,g\sim\rho}\left[\gamma\sum_{s',a'}p(s'|s,a)\pi(a'|s',g)S(s',a',g) - S(s,a,g)\right] \tag{45}$$

$$- D_f(\texttt{Mix}_\beta(d,\rho)(s,a,g) \,||\, \texttt{Mix}_\beta(q,\rho)(s,a,g)) \tag{46}$$

Now using the fact that strong duality holds in this problem we can swap the inner max and min resulting in:

$$= \max_{\pi}\min_{S(s,a,g)}\max_{\texttt{Mix}_\beta(d,\rho)(s,a,g)\geqslant 0}\beta(1-\gamma)\mathbb{E}_{d_0(s,g),\pi(a|s,g)}[S(s,a,g)]$$

$$+ \beta\mathbb{E}_{s,a,g\sim d}\left[\gamma\sum_{s',a'}p(s'|s,a)\pi(a'|s')S(s',a',g) - S(s,a,g)\right]$$

$$+ (1-\beta)\mathbb{E}_{s,a,g\sim\rho}\left[\gamma\sum_{s',a'}p(s'|s,a)\pi(a'|s')S(s',a',g) - S(s,a,g)\right]$$

$$- (1-\beta)\mathbb{E}_{s,a,g\sim\rho}\left[\gamma\sum_{s',a'}p(s'|s,a)\pi(a'|s',g)S(s',a',g) - S(s,a,g)\right] \tag{47}$$

$$- D_f(\texttt{Mix}_\beta(d,\rho)(s,a,g) \,||\, \texttt{Mix}_\beta(q,\rho)(s,a,g)) \tag{48}$$

$$\tag{49}$$

We can now apply the convex conjugate (Eq. (39)) definition to obtain a closed form for the inner maximization problem simplifying to:

$$\max_{\pi(a|s,g)}\min_{S(s,a,g)}\beta(1-\gamma)\mathbb{E}_{d_0(s,g),\pi(a|s,g)}[S(s,a,g)]$$

$$+ \mathbb{E}_{s,a,g\sim\texttt{Mix}_\beta(q,\rho)(s,a,g)}\left[f^*(\gamma\sum_{s',a'}p(s'|s,a,g)\pi(a'|s')S(s',a',g) - S(s,a,g))\right]$$

$$- (1-\beta)\mathbb{E}_{s,a,g\sim\rho}\left[\gamma\sum_{s',a'}p(s'|s,a,g)\pi(a'|s')S(s',a',g) - S(s,a,g)\right] \tag{50}$$

This completes our derivation of the SMORe objective. Since strong duality holds (objective convex, constraints linear and feasible), SMORe and the primal mixture occupancy matching share the same global optima $\pi_g^*$. $\qquad\square$

### A.4 ACTION-FREE SMORE: DUAL-V OBJECTIVE FOR OFFLINE GOAL CONDITIONED REINFORCEMENT LEARNING

The primal problem in Equation 6 is over-constrained. The objective determines the visitation distribution $d$ uniquely under a fixed policy. It turns out we can further relax this constraint to get an objective that results in the same optimal solution (Agarwal et al., 2019) $\pi_g^*$ by rewriting our primal formulation as:

$$\max_{d(s,a,g)\geqslant 0} -D_f(\texttt{Mix}_\beta(d,\rho)(s,a,g) \,||\, \texttt{Mix}_\beta(q,\rho)(s,a,g))$$

$$\text{s.t} \sum_a d(s,a,g) = (1-\gamma)\rho_0(s,g) + \gamma\sum_{s',a'}d(s',a',g)p(s|s',a'). \tag{51}$$

**Theorem 2.** *Let* $y(s,a,g) = \gamma\mathbb{E}_{s'\sim p(\cdot|s,a)}[S(s',g)] - S(s,g)$. *The action-free dual problem to the multi-task mixture occupancy matching objective (Equation 51) is given by:*

$$\min_{S(s,g)}\beta(1-\gamma)\mathbb{E}_{d_0(s,g)}[S(s,g)]$$

$$+\mathbb{E}_{s,a,g\sim\texttt{Mix}_\beta(q,\rho)(s,a,g)}\left[\max\left(0,(f')^{-1}(y(s,a,g))\right)y(s,a,g) - f\left(\max\left(0,(f')^{-1}(y(s,a,g))\right)\right)\right]$$

$$- (1-\beta)\mathbb{E}_{s,a,g\sim\rho}\left[\gamma\sum_{s'}p(s'|s,a)S(s',g) - S(s,g)\right]$$

*where $S$ is the lagrange dual variable defined as $S : \mathcal{S} \times \mathcal{G} \to \mathbb{R}$. Moreover, strong duality holds from Slater's conditions the primal and dual share the same optimal solution $\pi_g^*$ for any offline transition distribution $d^O$.*

*Proof.* Proceeding as before and applying Lagrangian duality and convex conjugate (39) to this problem, we can convert it to an unconstrained problem with dual variables $S(s,g)$ defined for all $s, g \in \mathcal{S} \times \mathcal{G}$:

$$\max_{d \geqslant 0} \min_{S(s,g)} -D_f(\texttt{Mix}_\beta(d,\rho)(s,a,g) \,||\, \texttt{Mix}_\beta(q,\rho)(s,a,g))$$

$$+ \sum_{s,g} S(s,g) \left( (1-\gamma)d_0(s,g) + \gamma \sum_{s',a',g} d(s',a',g)p(s|s',a',g) - \sum_a d(s,a,g) \right) \qquad (52)$$

$$= \max_{d \geqslant 0} \min_{S(s,g)} (1-\gamma)\mathbb{E}_{d_0(s,g)}[S(s,g)]$$

$$+ \mathbb{E}_{s,a,g \sim d}\left[ \gamma \sum_{s'} p(s'|s,a)\pi(a'|s')S(s',g) - S(s,g) \right] \qquad (53)$$

$$- D_f(\texttt{Mix}_\beta(d,\rho)(s,a,g) \,||\, \texttt{Mix}_\beta(q,\rho)(s,a,g)) \qquad (54)$$

$$= \max_{d \geqslant 0} \min_{S(s,g)} \beta(1-\gamma)\mathbb{E}_{d_0(s,g)}[S(s,g)]$$

$$+ \beta\mathbb{E}_{s,a,g \sim d}\left[ \gamma \sum_{s'} p(s'|s,a)S(s',g) - S(s,g) \right]$$

$$+ (1-\beta)\mathbb{E}_{s,a,g \sim d^O}\left[ \gamma \sum_{s'} p(s'|s,a)S(s',g) - S(s,g) \right]$$

$$- (1-\beta)\mathbb{E}_{s,a,g \sim d^O}\left[ \gamma \sum_{s'} p(s'|s,a)S(s',g) - S(s,g) \right] \qquad (55)$$

$$- D_f(\texttt{Mix}_\beta(d,\rho)(s,a,g) \,||\, \texttt{Mix}_\beta(q,\rho)(s,a,g)) \qquad (56)$$

Now using the fact that strong duality holds in this problem we can swap the inner max and min resulting in:

$$= \min_{S(s,g)} \max_{\texttt{Mix}_\beta(d,\rho)(s,a,g) \geqslant 0} \beta(1-\gamma)\mathbb{E}_{d_0(s,g)}[S(s,g)]$$

$$+ \beta\mathbb{E}_{s,a,g \sim d}\left[ \gamma \sum_{s'} p(s'|s,a)S(s',g) - S(s,g) \right]$$

$$+ (1-\beta)\mathbb{E}_{s,a,g \sim d^O}\left[ \gamma \sum_{s'} p(s'|s,a)S(s',g) - S(s,g) \right]$$

$$- (1-\beta)\mathbb{E}_{s,a,g \sim d^O}\left[ \gamma \sum_{s'} p(s'|s,a)S(s',g) - S(s,g) \right] \qquad (57)$$

$$- D_f(\texttt{Mix}_\beta(d,\rho)(s,a,g) \,||\, \texttt{Mix}_\beta(q,\rho)(s,a,g)) \qquad (58)$$

Unlike previous case where constraints uniquely define a valid $d$ for any given $\pi$, in this case we need to take into account the hidden constraint $d \geqslant 0$ or equivalently $\texttt{Mix}_\beta(d,\rho)(s,a,g) \geqslant 0$. To incorporate the non-negativity constraints we consider the inner maximization separately and derive a closed-form solution that adheres to the non-negativity constraints. Recall $y(s,a,g) = \mathbb{E}_{s' \sim p(s,a)}[S(s',g)] - S(s,g)$.

$$\max_{\mathtt{Mix}_\beta(d,\rho)(s,a,g) \geqslant 0} \mathbb{E}_{s,a,g \sim \mathtt{Mix}_\beta(d,\rho)(s,a,g)} \left[ \gamma \sum_{s'} p(s'|s,a) S(s',g) - S(s,g) \right]$$
$$- D_f(\mathtt{Mix}_\beta(d,\rho)(s,a,g) \,\|\, \mathtt{Mix}_\beta(q,\rho)(s,a,g))$$

We can now construct the Lagrangian dual to incorporate the constraint $\mathtt{Mix}_\beta(d,\rho)(s,a,g) \geqslant 0$ in its equivalent form $w(s,a,g) \geqslant 0$ and obtain the following where $w \triangleq \frac{\mathtt{Mix}_\beta(d,\rho)(s,a,g)}{\mathtt{Mix}_\beta(q,\rho)(s,a,g)}$:

$$\max_{w(s,a,g)} \max_{\lambda \geqslant 0} \mathbb{E}_{s,a \sim \mathtt{Mix}_\beta(q,\rho)(s,a,g)}[w(s,a,g)y(s,a,g)] - \mathbb{E}_{\mathtt{Mix}_\beta(q,\rho)(s,a,g)}[f(w(s,a,g))] + \sum_{s,a,g} \lambda(w(s,a,g) - 0)$$

(59)

Since strong duality holds, we can use the KKT constraints to find the solutions $w^*(s,a,g)$ and $\lambda^*(s,a,g)$.

1. **Primal feasibility**: $w^*(s,a,g) \geqslant 0 \ \ \forall \, s,a$

2. **Dual feasibility**: $\lambda^* \geqslant 0 \ \ \forall \, s,a$

3. **Stationarity**: $\mathtt{Mix}_\beta(q,\rho)(s,a,g)(-f'(w^*(s,a,g)) + y(s,a,g) + \lambda^*(s,a,g)) = 0 \ \ \forall \, s,a$

4. **Complementary Slackness**: $(w^*(s,a,g) - 0)\lambda^*(s,a,g) = 0 \ \ \forall \, s,a$

Using stationarity we have the following:
$$f'(w^*(s,a,g)) = y(s,a,g) + \lambda^*(s,a,g) \ \ \forall \, s,a,g \tag{60}$$
Now using complementary slackness, only two cases are possible $w^*(s,a,g) \geqslant 0$ or $\lambda^*(s,a,g) \geqslant 0$. Combining both cases we arrive at the following solution for this constrained optimization:
$$w^*(s,a) = \max\left(0, f'^{-1}(y(s,a,g))\right) \tag{61}$$

Using the optimal closed-form solution ($w^*$) for $\mathtt{Mix}_\beta(d,\rho)(s,a,g)$ of the inner optimization in Eq. (57) we obtain

$$\min_{S(s,a)} \beta(1-\gamma)\mathbb{E}_{d_0(s)}[S(s,g)]$$
$$+ \mathbb{E}_{s,a,g \sim \mathtt{Mix}_\beta(q,\rho)(s,a,g)}\left[\max\left(0, (f')^{-1}(y(s,a,g))\right) y(s,a,g) - \alpha f\left(\max\left(0, (f')^{-1}(y(s,a,g))\right)\right)\right]$$
$$- (1-\alpha)\mathbb{E}_{s,a \sim \rho}\left[\gamma \sum_{s'} p(s'|s,a)\pi(a'|s') S(s',g) - S(s,g)\right] \tag{62}$$

For deterministic dynamics, this reduces to the action-free SMORe objective:
$$\min_{S(s,a)} \beta(1-\gamma)\mathbb{E}_{d_0(s)}[S(s,g)]$$
$$+ \mathbb{E}_{s,a \sim \mathtt{Mix}_\beta(q,\rho)(s,a,g)}\left[\max\left(0, (f')^{-1}(y(s,a,g))\right) y(s,a,g) - f\left(\max\left(0, (f')^{-1}(y(s,a,g))\right)\right)\right]$$
$$- (1-\beta)\mathbb{E}_{s,a \sim \rho}\left[\gamma S(s',g) - S(s,g)\right] \tag{63}$$

where $y(s,a,g) = \gamma S(s',g) - S(s,g)$.

Note that we no longer need actions in the offline dataset to learn an optimal goal conditioned score function. This score function can be used to learn presentation in action-free datasets as well as for transfer of value function across differing action-modalities where agents share the same observation space (eg. images as observations).

$\square$

## B   SMORE ALGORITHMIC DETAILS

### B.1   SMORE WITH COMMON $f$-DIVERGENCES

**a. KL divergence**

We consider the reverse KL divergence and start with the general SMORe objective:

$$\max_{\pi_g} \min_{S} \beta(1-\gamma)\mathbb{E}_{d_0,\pi_g}[S(s,a,g)] + \mathbb{E}_{s,a,g\sim\text{Mix}_\beta(q,\rho)(s,a,g)}[f^*(\gamma P^{\pi_g}S(s,a,g) - S(s,a,g))]$$
$$- (1-\beta)\mathbb{E}_{s,a,g\sim\rho}[\gamma P^{\pi_g}S(s,a,g) - S(s,a,g)] \quad (64)$$

Plugging in the conjugate $f^*$ for reverse KL divergence we get:

$$\max_{\pi_g} \min_{S} \beta(1-\gamma)\mathbb{E}_{d_0,\pi_g}[S(s,a,g)] + \mathbb{E}_{s,a,g\sim\text{Mix}_\beta(q,\rho)(s,a,g)}\Big[e^{(\gamma P^{\pi_g}S(s,a,g) - S(s,a,g))}\Big]$$
$$- (1-\beta)\mathbb{E}_{s,a,g\sim\rho}[\gamma P^{\pi_g}S(s,a,g) - S(s,a,g)] \quad (65)$$

Using the telescoping sum for the last term in the objective above, we can simplify it as follows:

$$\max_{\pi_g} \min_{S} \beta(1-\gamma)\mathbb{E}_{d_0,\pi_g}[S(s,a,g)] + \mathbb{E}_{s,a,g\sim\text{Mix}_\beta(q,\rho)(s,a,g)}\Big[e^{(\gamma P^{\pi_g}S(s,a,g) - S(s,a,g))}\Big]$$
$$+ (1-\beta)\mathbb{E}_{s,g\sim d_0,a\sim\rho(\cdot|s,g)}[S(s,a,g)] \quad (66)$$

With the initial state distribution $d_0$ set to the offline dataset distribution $\rho$, and Since our initial state distribution is the same as offline data distribution, we get:

$$\max_{\pi_g} \min_{S} \beta(1-\gamma)\mathbb{E}_{\rho,\pi_g}[S(s,a,g)] + \mathbb{E}_{s,a,g\sim\text{Mix}_\beta(q,\rho)(s,a,g)}\Big[e^{(\gamma P^{\pi_g}S(s,a,g) - S(s,a,g))}\Big]$$
$$+ (1-\beta)\mathbb{E}_\rho[S(s,a,g)] \quad (67)$$

Collecting terms together we get:

$$\max_{\pi_g} \min_{Q} \mathbb{E}_\rho[\mathbb{E}_{a\sim\pi}[\beta(1-\gamma)S(s,a,g)] + \mathbb{E}_{a\sim\rho}[(1-\beta)S(s,a,g)]]$$
$$+ \mathbb{E}_{s,a,g\sim\text{Mix}_\beta(q,\rho)(s,a,g)}\Big[e^{(\gamma P^{\pi_g}S(s,a,g) - S(s,a,g))}\Big] \quad (68)$$

The objective for SMORe with reverse KL divergence pushes down the "score" of offline dataset transitions selectively (without pushing down score of the goal-transition distribution) while minimizing the term resembling bellman regularization that also encourages increasing score at the mixture dataset jointly over the offline dataset as well as the goal transition distribution.

**b. Pearson chi-squared divergence**

We consider the Pearson $\chi^2$ and start with the general SMORe objective:

$$\max_{\pi_g} \min_{S} \beta(1-\gamma)\mathbb{E}_{d_0,\pi_g}[S(s,a,g)] + \mathbb{E}_{s,a,g\sim\text{Mix}_\beta(q,\rho)(s,a,g)}[f^*(\gamma P^{\pi_g}S(s,a,g) - S(s,a,g))]$$
$$- (1-\beta)\mathbb{E}_{s,a,g\sim\rho}[\gamma P^{\pi_g}S(s,a,g) - S(s,a,g)] \quad (69)$$

With the initial state distribution $d_0$ set to the offline dataset distribution $\rho$, and plugging in the conjugate $f^*$ for Pearson $\chi^2$ divergence we get:

$$\max_{\pi_g} \min_{S} \beta(1-\gamma)\mathbb{E}_{d_0,\pi_g}[S(s,a,g)] + 0.25\mathbb{E}_{s,a,g\sim\text{Mix}_\beta(q,\rho)(s,a,g)}\Big[(\gamma P^{\pi_g}S(s,a,g) - S(s,a,g))^2\Big]$$
$$+ \mathbb{E}_{s,a,g\sim\text{Mix}_\beta(q,\rho)(s,a,g)}[(\gamma P^{\pi_g}S(s,a,g) - S(s,a,g))] - (1-\beta)\mathbb{E}_{s,a,g\sim\rho}[\gamma P^{\pi_g}S(s,a,g) - S(s,a,g)]$$
$$(70)$$

Using the fact that $\mathtt{Mix}_\beta(q,\rho)(s,a,g) = \beta q(s,a,g) + (1-\beta)\rho(s,a,g)$, we can further simplify the above equation to:

$$\max_{\pi_g} \min_{S} \beta(1-\gamma)\mathbb{E}_{d_0,\pi_g}[S(s,a,g)] + 0.25\mathbb{E}_{s,a,g\sim\mathtt{Mix}_\beta(q,\rho)(s,a,g)}\left[(\gamma P^{\pi_g}S(s,a,g) - S(s,a,g))^2\right]$$
$$+ \beta\mathbb{E}_{s,a,g\sim q}[(\gamma P^{\pi_g}S(s,a,g) - S(s,a,g))] \quad (71)$$

Collecting terms together we get:

$$\max_{\pi_g} \min_{S} \beta(1-\gamma)\mathbb{E}_{\rho,\pi_g}[S(s,a,g)] + \beta\mathbb{E}_{s,g\sim q,a\sim\pi_g}[\gamma P^{\pi_g}S(s,a,g)]$$
$$- \beta\mathbb{E}_{s,a,g\sim q}[S(s,a,g)] + 0.25\mathbb{E}_{s,a,g\sim\mathtt{Mix}_\beta(q,\rho)(s,a,g)}\left[(\gamma P^{\pi_g}S(s,a,g) - S(s,a,g))^2\right] \quad (72)$$

Observing the equation above, we note that the first two terms decrease score at offline data distribution as well as the goal transition distribution when actions are sampled according to the policy $\pi_g$. Simultaneously the third term pushes score up for the $\{s,a,g\}$ tuples that are sampled from goal transition distribution. Finally the last term encouraged enforces a bellman regularization enforcing smoothness is the scores of neighbouring states.

## C  SMORE EXPERIMENTAL DETAILS

### C.1  TASKS WITH OBSERVATIONS AS STATES

**Environments:** For the offline GCRL experiments we consider the benchmark used in prior work GoFar and extend it with locomotion tasks. For the manipulations tasks we consider the Fetch environment and a dextrous shadow hand environment. Fetch environments (Plappert et al., 2018) consists of a manipulator with seven degrees of freedom along with a parallel gripper. The set of environments get a sparse reward of 1 when the goal is within 5 cm and 0 otherwise. The action space is 4 dimensional (3 dimension cartesian control + 1 dimension gripper control). The shadow hand is 24 DOF manipulator with 20-dimensional action space. The goal is 15-dimension specifying the position for each of the five fingers. The tolerance for goal reaching is 1 cm. For the locomotion environments, the task is to achieve a particular velocity in the x direction and stay at the velocity. For HalfCheetah, the target velocity is set to 11.0 and for Ant the target velocity is 5.0. For locomotion environments, the tolerance for goal reaching if 0.5. The MuJoCo environments used in this work are licensed under CC BY 4.0.

**Offline Datasets:** We use existing datasets from the offline GCRL benchmark used in (Ma et al., 2022) for all manipulation tasks except Reacher, SawyerReach, and SawyerDoor. For Reacher, SawyerReach, and SawyerDoor we use existing datasets from (Yang et al., 2022). These datasets are comprised on x% random data and (100-x)% expert data depending on the coverage over goals reached in individual datasets. We create our own datasets for locomotion by using 'random/medium/medium-replay' data as our offline (suboptimal) data combined with 30 trajectories from corresponding 'expert' datasets. The datasets used from D4RL are licensed under Apache 2.0.

**Baselines:** To benchmark and analyze the performance of our proposed methods for offline imitation learning with suboptimal data, we consider the following representative baselines in this work: GoFAR (Ma et al., 2022), WGCSL (Yang et al., 2022), GCSL (Ghosh et al., 2019), and Actionable Models (Chebotar et al., 2021), Contrastive RL (Eysenbach et al., 2020) and GC-IQL Kostrikov et al. (2021). GoFAR is a dual occupancy matching approach to GCRL that formulates it as a weighted regression problem. WGCSL and GSCL use goal-conditioned behavior cloning with goal relabelling as the base algorithms and WGCL uses weights to learn improved policy over GCSL. Actionable models uses conservative learning with goal chaining to learn goal-reaching behaviours using offline datasets. Contrastive RL treats GCRL as a classification problem - contrastive goals that are achieved in trajectory from random goals. Finally, GC-IQL extends the single task offline RL algorithm IQL to GCRL.

The open-source implementations of the baselines GoFAR, WGCSL, GCSL, Actionable models, Contrastive RL and IQL are provided by the authors (Ma et al., 2022) and employed in our experiments. We use the hyperparameters provided by the authors, which are consistent with those used in the original GoFAR paper, for all the MuJoCo locomotion and manipulation environments. We implement

contrastive learning using the code from Contrastive RL repository. GC-IQL is implemented using code from author's implementation found here.

**Architecture and Hyperparameters** For the baselines, we use tuned hyperparameters from previous works that were tuned on the same set of tasks and datasets. Implementation for `SMORe` shares the same network architecture as baselines. GoFAR additionally requires training a discriminator. For all experiments, all methods are trained for 10 seeds with each training run. Fetch manipulation (except Push) tasks are trained for a maximum of 400k minibatch updates of size 512 whereas all other environments training is done for 1M minibatch updates. The expectile parameter $\tau$ was searched over [0.65, 0.7, 0.8, 0.85]. For the results shown in table 1, Fetch and Sawyer environments use $\tau = 0.8$, Locomotion and Adroit hand environments use $\tau = 0.7$. In general, the HER ratio is searched over [0.2, 0.5, 0.8, 1.0] for all methods and the best one was selected. HER ratio of 0.8 gave best performance across all tasks for `SMORe`.

The architectures and hyperparameters for all methods are reported in Table 4.

| Hyperparameter | Value |
|---|---|
| Policy updates $n_{pol}$ | 1 |
| Policy learning rate | 3e-4 |
| Value learning rate | 3e-4 |
| MLP layers | (256,256) |
| LR decay schedule | cosine |
| Discount factor | 0.99 |
| LR decay schedule | cosine |
| Batch Size | 512 |
| Mixture ratio $\beta$ | 0.5 |
| Policy temperature ($\alpha$) | 3.0 |

Table 4: Hyperparameters for `SMORe`.

## C.2 TASKS WITH OBSERVATIONS AS IMAGES

| Hyperparameters | Values |
|---|---|
| batch size | 2048 |
| number of training epochs | 300 |
| number of training iterations per epoch | 1000 |
| Horizon | 400 |
| image encoder architecture | 3-layer CNN |
| policy network architecture | (1024, 4) MLP |
| critic/score network architecture | (1024, 4) MLP |
| weight initialization for final layers of critic and policy | $\text{UNIF}[-10^{-12}, 10^{-12}]$ |
| policy std deviation | 0.15 |
| representation dimension | 16 |
| data augmentation | random cropping |
| discount factor | 0.99 |
| learning rate | 3e-4 |

Table 5: Hyperparameters for image-observation GCRL from Zheng et al. (2023).

**Tasks and dataset** Our experiments use a suite of simulated goal-conditioned control tasks based on prior work Zheng et al. (2023). The observations and goals are $48 \times 48 \times 3$ RGB images. The evaluation tasks require multi-trajectory stitching whereas the dataset contains trajectories solving only parts of the evaluation tasks.

| Task | Behavior cloning | | Contrastive RL | RL+sparse reward | |
|------|-----------------|---|----------------|-----------------|---|
| | WGCSL | GCSL | CRL | AM | IQL |
| Reacher | $15.30 \pm_{0.58}$ | $14.01 \pm_{0.36}$ | $16.62 \pm_{2.09}$ | $23.68 \pm_{0.58}$ | $8.86 \pm_{0.61}$ |
| SawyerReach | $14.06 \pm_{0.08}$ | $12.05 \pm_{1.23}$ | $23.03 \pm_{1.17}$ | $23.37 \pm_{2.29}$ | $36.19 \pm_{0.01}$ |
| SawyerDoor | $16.79 \pm_{0.75}$ | $18.29 \pm_{0.94}$ | $12.26 \pm_{3.94}$ | $16.63 \pm_{0.76}$ | $29.31 \pm_{0.88}$ |
| FetchPick | $6.87 \pm_{0.77}$ | $6.54 \pm_{1.85}$ | $0.21 \pm_{0.29}$ | $0.45 \pm_{0.32}$ | $15.24 \pm_{1.27}$ |
| FetchPush | $10.62 \pm_{0.98}$ | $12.38 \pm_{1.10}$ | $3.60 \pm_{0.59}$ | $2.74 \pm_{0.70}$ | $19.95 \pm_{1.94}$ |
| FetchSlide | $2.62 \pm_{1.15}$ | $2.03 \pm_{0.01}$ | $0.41 \pm_{0.03}$ | $0.31 \pm 0.31$ | $3.25 \pm_{1.02}$ |

Table 6: Discounted Return for the offline GCRL benchmark with 5% expert data. Results are averaged over 10 seeds.

| Task | Behavior cloning | | Contrastive RL | RL+sparse reward | |
|------|-----------------|---|----------------|-----------------|---|
| | WGCSL | GCSL | CRL | AM | IQL |
| Reacher | $13.03 \pm_{0.56}$ | $12.17 \pm_{0.8}$ | $19.63 \pm_{3.09}$ | $24.78 \pm_{0.23}$ | $4.44 \pm_{0.70}$ |
| SawyerReach | $11.455 \pm_{1.37}$ | $11.34 \pm_{1.18}$ | $25.35 \pm_{0.8}$ | $25.19 \pm_{0.61}$ | $35.73 \pm_{0.22}$ |
| SawyerDoor | $16.79 \pm_{0.29}$ | $13.20 \pm_{0.53}$ | $14.78 \pm_{5.29}$ | $16.59 \pm_{1.39}$ | $16.87 \pm_{4.21}$ |
| FetchPick | $4.39 \pm_{1.35}$ | $4.99 \pm_{0.11}$ | $0.21 \pm_{0.29}$ | $0.24 \pm_{0.27}$ | $11.79 \pm_{1.78}$ |
| FetchPush | $8.01 \pm_{1.96}$ | $8.04 \pm_{0.34}$ | $3.60 \pm_{0.59}$ | $2.02 \pm_{0.48}$ | $19.66 \pm_{1.69}$ |
| FetchSlide | $2.33 \pm_{0.23}$ | $2.37 \pm_{0.83}$ | $0.44 \pm_{0.016}$ | $0.45 \pm_{0.44}$ | $1.83 \pm_{1.31}$ |

Table 7: Discounted Return for the offline GCRL benchmark with 2.5% expert data. Results are averaged over 10 seeds.

In the simulation, we employed an offline manipulation dataset comprising near-optimal examples of basic action sequences, including the initiation of drawer movement, the displacement of blocks, and the grasping of items. The demonstrations exhibit variations in length, ranging between 50 to 100 horizon, while the offline dataset contains a total of 250,000 state transitions in its entirety. It is important to note that the offline trajectories do not depict a complete progression from the initial condition to the final objective. For the purposes of evaluation, we consider 4 tasks similar to Zheng et al. (2023), against which we compare the success rates in realizing these specified objectives.

**Baseline and `SMORe` implementations.** We use the open-source implementation of Stable-contrastive RL to use the same design decisions and implement `SMORe`, GC-IQL on that codebase. We use the same hyperparameters as the stable-contrastive RL implementation for the shared hyperparameters. The hyperparameters for `SMORe` were kept the same as in Table 4.

# D    ADDITIONAL EXPERIMENTS

## D.1    RESULTS ON OFFLINE GCRL BENCHMARK WITH VARYING EXPERT COVERAGE IN OFFLINE DATASET

We ablate the effect of dataset quality on the performance of an offline GCRL method in this sections. Table 6, 7, 8 show performance of all methods with 5%, 2.5% and 1% expert data in the offline dataset respectively.

## D.2    SUCCESS RATE AND FINAL DISTANCE TO GOAL ON MANIPULATION TASKS

Table 10 and Table 11 reports the success rate and final distance to goal metrics on manipulation tasks.

## D.3    ROBUSTNESS OF MIXTURE DISTRIBUTION PARAMETER $\beta$

We find that `SMORe` is quite robust to the mixture distribution parameter $\beta$ except in the environment FetchPush where $\beta = 0.5$ is the most performant. Table 9 shows this result empirically.

| Task | Behavior cloning | | Contrastive RL | RL+sparse reward | |
|---|---|---|---|---|---|
| | WGCSL | GCSL | CRL | AM | IQL |
| Reacher | $13.56_{\pm 0.69}$ | $12.27_{\pm 1.45}$ | $17.94_{\pm 3.71}$ | $24.89_{\pm 0.34}$ | $4.28 \pm 0.92$ |
| SawyerReach | $10.71 \pm_{0.69}$ | $11.79_{\pm 1.46}$ | $25.61_{\pm 0.39}$ | $25.54_{\pm 0.95}$ | $31.31 \pm 2.08$ |
| SawyerDoor | $15.18_{\pm 0.81}$ | $11.89_{\pm 1.51}$ | $10.26_{\pm 4.61}$ | $18.04_{\pm 1.8}$ | $17.11 \pm 4.45$ |
| FetchPick | $1.89 \pm 1.22$ | $3.30 \pm 0.66$ | $0.42 \pm 0.29$ | $0.41 \pm 0.22$ | $7.90 \pm 1.22$ |
| FetchPush | $6.44 \pm 3.64$ | $6.43 \pm 0.56$ | $1.69 \pm 1.56$ | $2.63 \pm 3.04$ | $7.11 \pm 2.60$ |
| FetchSlide | $1.77 \pm 0.24$ | $1.11 \pm 0.26$ | $0.0 \pm 0.0$ | $0.10 \pm 0.11$ | $0.80 \pm 0.48$ |

Table 8: Discounted Return for the offline GCRL benchmark with 1% expert data. Results are averaged over 10 seeds.

## D.4 HOW MUCH DOES HER CONTRIBUTE TO THE PERFORMANCE IMPROVEMENTS OF SMORE?

GoFAR demonstrated improved performance without relying on HER. The authors also demonstrated that HER is detrimental to GoFAR's performance. In this section, we aim to conduct a similar stufy and see how much HER contributed to SMORe's performance. Table 12 shows that HER gives SMORe a small performance boost and show that SMORe is still able to outperform GoFAR without HER.

## D.5 COMPARISON TO VARIANTS OF GOFAR

GoFAR formulates GCRL as an occupancy matching problem, but it is also suggested that using a discriminator is optional. Without a discriminator, GoFAR reduces to a sparse reward RL problem. Table 13 shows that GoFAR achieves poor performance when a reward function is substituted in place on an discriminator. We also study if the performance benefits we obtain are due to the offline learning strategy we used from IQL. We modify GoFAR with discriminator reward to use expectile loss for value learning and AWR for policy learning. Results in Table 13 shows that no performance gains were observed.

## D.6 OFFLINE GCRL WITH PURELY SUBOPTIMAL DATA

In this experiment, we study offline GCRL from purely suboptimal datasets. Except FetchReach, these datasets provide very sparse coverage of goals expected to reach in evaluation. Table 14 shows the robustness of SMORe even in the setting of poor quality offline data.

## D.7 COMPARISON WITH IN-SAMPLE LEARNING METHODS

In-sample learning methods perform value improvement using bellman backups without OOD action sampling. This makes them a particularly suitable candidate for offline setting. We compare against a number of recent in-sample learning methods, IQL (Kostrikov et al., 2021), SQL/EQL Xu et al. (2023) and XQL (Garg et al., 2023). Table 15 compares SMORe to in-sample learning methods adapted to GCRL.

## D.8 ABLATING COMPONENTS OF SMORE FOR OFFLINE SETTING

In offline setting, it is well known that bellman backups suffer from overestimation and results in poor policy performance. We validate the utility of the components used in this work in Table 16 : expectile loss function and constrained policy optimization with AWR.

| Task | $\beta = 0.5$ | $\beta = 0.7$ | $\beta = 0.8$ | $\beta = 0.9$ |
|------|------|------|------|------|
| FetchReach | $35.08 \pm 0.54$ | $36.57 \pm 0.20$ | $36.59 \pm 0.30$ | $36.30 \pm 0.30$ |
| FetchPick | $26.47 \pm 0.34$ | $27.04 \pm 0.81$ | $27.43 \pm 0.97$ | $27.89 \pm 1.19$ |
| FetchPush | $26.83 \pm 1.21$ | $16.20 \pm 1.11$ | $11.50 \pm 1.19$ | $13.85 \pm 5.53$ |
| FetchSlide | $4.99 \pm 0.40$ | $3.76 \pm 0.75$ | $3.43 \pm 2.4$ | $4.10 \pm 1.20$ |

Table 9: Discounted Return for the offline GCRL benchmark with varying mixture coefficients in offline dataset. Results are averaged over 10 seeds.

| Task | Occupancy Matching | | Behavior cloning | | Contrastive RL | RL+sparse reward | |
| | SMORe | GoFAR | WGCSL | GCSL | CRL | AM | IQL |
|------|------|------|------|------|------|------|------|
| Reacher | $0.875 \pm 0.07$ | $0.90 \pm 0.01$ | $0.97 \pm 0.014$ | $0.92 \pm 0.08$ | $0.76 \pm 0.74$ | $1.0 \pm 0.1$ | $0.26 \pm 0.06$ |
| SawyerReach | $0.98 \pm 0.014$ | $0.75 \pm 0.04$ | $1.0 \pm 0.0$ | $0.98 \pm 0.02$ | $0.98 \pm 0.018$ | $1.0 \pm 0.1$ | $0.81 \pm 0.01$ |
| SawyerDoor | $0.875 \pm 0.038$ | $0.5 \pm 0.12$ | $0.78 \pm 0.10$ | $0.5 \pm 0.12$ | $0.22 \pm 0.11$ | $0.3 \pm 0.11$ | $0.84 \pm 0.06$ |
| FetchReach | $1.0 \pm 0.0$ | $1.0 \pm 0.0$ | $1.0 \pm 0.0$ | $0.98 \pm 0.05$ | $1.0 \pm 0.0$ | $1.0 \pm 1.0$ | $1.0 \pm 0.0$ |
| FetchPick | $0.925 \pm 0.045$ | $0.84 \pm 0.09$ | $0.54 \pm 0.16$ | $0.54 \pm 0.20$ | $0.42 \pm 0.29$ | $0.78 \pm 0.15$ | $0.86 \pm 0.11$ |
| FetchPush | $0.90 \pm 0.07$ | $0.88 \pm 0.09$ | $0.76 \pm 0.12$ | $0.72 \pm 0.15$ | $0.06 \pm 0.03$ | $0.67 \pm 0.14$ | $0.65 \pm 0.052$ |
| FetchSlide | $0.315 \pm 0.07$ | $0.18 \pm 0.12$ | $0.18 \pm 0.14$ | $0.17 \pm 0.13$ | $0.0 \pm 0.0$ | $0.11 \pm 0.09$ | $0.26 \pm 0.057$ |
| HandReach | $0.47 \pm 0.11$ | $0.40 \pm 0.20$ | $0.25 \pm 0.23$ | $0.047 \pm 0.10$ | $0.0 \pm 0.0$ | $0.0 \pm 0.0$ | $0.0 \pm 0.0$ |

Table 10: Success Rate for the offline GCRL benchmark with 10% expert data. Results are averaged over 10 seeds.

| Task | Occupancy Matching | | Behavior cloning | | Contrastive RL | RL+sparse reward | |
| | SMORe | GoFAR | WGCSL | GCSL | CRL | AM | IQL |
|------|------|------|------|------|------|------|------|
| Reacher | $0.02 \pm 0.01$ | $0.03 \pm 0.01$ | $0.011 \pm 0.01$ | $0.016 \pm 0.00$ | $0.05 \pm 0.03$ | $0.013 \pm 0.00$ | $0.12 \pm 0.005$ |
| SawyerReach | $0.008 \pm 0.004$ | $0.04 \pm 0.00$ | $0.004 \pm 0.00$ | $0.00 \pm 0.00$ | $0.01 \pm 0.01$ | $0.01 \pm 0.00$ | $0.053 \pm 0.004$ |
| SawyerDoor | $0.02 \pm 0.029$ | $0.18 \pm 0.00$ | $0.011 \pm 0.00$ | $0.017 \pm 0.01$ | $0.14 \pm 0.07$ | $0.06 \pm 0.01$ | $0.019 \pm 0.01$ |
| FetchReach | $0.004 \pm 0.0012$ | $0.018 \pm 0.003$ | $0.007 \pm 0.0043$ | $0.008 \pm 0.008$ | $0.007 \pm 0.001$ | $0.007 \pm 0.001$ | $0.002 \pm 0.001$ |
| FetchPick | $0.04 \pm 0.018$ | $0.036 \pm 0.013$ | $0.094 \pm 0.043$ | $0.108 \pm 0.06$ | $0.25 \pm 0.025$ | $0.04 \pm 0.02$ | $0.04 \pm 0.012$ |
| FetchPush | $0.03 \pm 0.003$ | $0.033 \pm 0.008$ | $0.041 \pm 0.02$ | $0.042 \pm 0.018$ | $0.15 \pm 0.036$ | $0.07 \pm 0.039$ | $0.05 \pm 0.006$ |
| FetchSlide | $0.09 \pm 0.012$ | $0.12 \pm 0.02$ | $0.173 \pm 0.04$ | $0.204 \pm 0.051$ | $0.42 \pm 0.05$ | $0.198 \pm 0.059$ | $0.09 \pm 0.013$ |
| HandReach | $0.039 \pm 0.0108$ | $0.024 \pm 0.009$ | $0.035 \pm 0.012$ | $0.038 \pm 0.013$ | $0.04 \pm 0.005$ | $0.037 \pm 0.004$ | $0.08 \pm 0.005$ |

Table 11: Final distance to goal for the offline GCRL benchmark with 10% expert data. Results are averaged over 10 seeds.

| Task | SMORe | SMORe w/o HER | GoFAR |
|------|------|------|------|
| FetchReach | $35.08 \pm 0.54$ | $34.86 \pm 1.03$ | $28.2 \pm 0.61$ |
| SawyerReach | $37.67 \pm 0.12$ | $37.34 \pm 0.36$ | $15.34 \pm 0.64$ |
| SawyerDoor | $31.48 \pm 0.46$ | $31.53 \pm 0.62$ | $18.94 \pm 0.01$ |
| FetchPick | $26.47 \pm 0.34$ | $25.72 \pm 3.88$ | $19.7 \pm 2.57$ |
| FetchPush | $26.83 \pm 1.21$ | $25.62 \pm 1.67$ | $18.2 \pm 3.00$ |
| FetchSlide | $4.99 \pm 0.40$ | $4.09 \pm 0.33$ | $2.47 \pm 1.44$ |

Table 12: Performance gains using HER (resampling ratio=0.8) on `SMORe`. All other hyperparameters are kept the same between SMORe and SMORe w/o HER.

| Task | SMORe | GoFAR (discriminator) | GoFAR (sparse reward) | r-GoFAR (sparse reward) | GoFAR (expectile loss+AWR) |
|------|------|------|------|------|------|
| FetchReach | $35.08 \pm 0.54$ | $28.2 \pm 0.61$ | $26.1 \pm 1.14$ | $0.30 \pm 0.43$ | $26.90 \pm 0.41$ |
| SawyerReach | $37.67 \pm 0.12$ | $15.34 \pm 0.64$ | — | $0.34 \pm 0.33$ | $16.17 \pm 3.02$ |
| SawyerDoor | $31.48 \pm 0.46$ | $18.94 \pm 0.01$ | — | $10.36 \pm 3.27$ | $22.47 \pm 1.13$ |
| FetchPick | $26.47 \pm 0.34$ | $19.7 \pm 2.57$ | $17.4 \pm 1.78$ | $0.25 \pm 0.02$ | $18.46 \pm 2.72$ |
| FetchPush | $26.83 \pm 1.21$ | $18.2 \pm 3.00$ | $17.4 \pm 2.67$ | $4.23 \pm 3.96$ | $17.39 \pm 5.44$ |
| FetchSlide | $4.99 \pm 0.40$ | $2.47 \pm 1.44$ | $5.13 \pm 4.05$ | $0.29 \pm 0.03$ | $3.59 \pm 2.30$ |

Table 13: Ablation comparison with GoFAR: a) a sparse binary reward is used in place of a learned discriminator in GoFAR b) The policy and value update is replaced by AWR and expectile loss respectively. The main difference with SMORe remains the use of discriminator for training. — indicates these environments were not considered in (Ma et al., 2022). Our reproduced results (denoted by r-) with the official code base for binary results did not match up to the reported results.

| Task | SMORe | GoFAR | WGCSL | GC-IQL |
|------|-------|-------|-------|--------|
| FetchReach | $35.01\pm_{0.47}$ | $27.37\pm_{0.4}$ | $21.65\pm_{0.61}$ | $23.72\pm_{1.18}$ |
| SawyerReach | $36.26\pm_{0.93}$ | $5.89\pm_{1.36}$ | $7.27\pm_{1.14}$ | $33.08\pm_{0.81}$ |
| SawyerDoor | $20.28\pm_{2.65}$ | $15.33\pm_{1.30}$ | $13.81\pm_{2.72}$ | $16.05\pm_{4.97}$ |
| FetchPick | $0.61\pm_{0.5}$ | $0.0\pm_{0.0}$ | $0.0\pm_{0.0}$ | $1.31\pm_{1.86}$ |
| FetchPush | $6.39\pm_{0.68}$ | $4.23\pm_{3.96}$ | $4.27\pm_{3.9}$ | $2.63\pm_{1.68}$ |
| FetchSlide | $0.42\pm_{0.01}$ | $0.059\pm_{0.08}$ | $0.93\pm_{0.69}$ | $0.75\pm_{0.58}$ |

Table 14: Discounted Return for the offline GCRL benchmark with 0% expert data. Results are averaged over 10 seeds.

| Task | SMORe | GC-IQL | GC-SQL | GC-EQL | GC-XQL |
|------|-------|--------|--------|--------|--------|
| FetchReach | $35.08\pm_{0.54}$ | $34.43\pm_{1.00}$ | $35.67\pm_{0.70}$ | $29.23\pm_{0.2}$ | $33.94\pm_{0.49}$ |
| SawyerReach | $37.67\pm_{0.12}$ | $35.18\pm_{0.29}$ | $37.10\pm_{0.24}$ | $30.19\pm_{1.66}$ | $32.88\pm_{2.85}$ |
| SawyerDoor | $31.48\pm_{0.46}$ | $25.52\pm_{1.45}$ | $27.96\pm_{0.45}$ | $3.57\pm_{3.51}$ | $5.85\pm_{4.21}$ |
| FetchPick | $26.47\pm_{0.34}$ | $16.8\pm_{3.10}$ | $18.35\pm_{6.67}$ | $1.31\pm_{1.86}$ | $1.31\pm_{1.82}$ |
| FetchPush | $26.83\pm_{1.21}$ | $22.40\pm_{0.74}$ | $17.19\pm_{2.56}$ | $2.64\pm_{1.30}$ | $3.79\pm_{0.21}$ |
| FetchSlide | $4.99\pm_{0.40}$ | $4.80\pm_{1.59}$ | $4.68\pm_{3.32}$ | $0.06\pm_{0.08}$ | $0.36\pm_{0.52}$ |

Table 15: Comparison of SMORe to in-sample RL methods - IQL (Kostrikov et al., 2021),SQL/EQL (Xu et al., 2023), XQL (Garg et al., 2023) that learn from sparse rewards. — in EQL denotes the learning diverged. We observed IQL to be the most stable alternative compared to SQL, EQL and XQL. SQL, EQL and XQL were implemented using author's official codebase

| Task | SMORe | SMORe (w/o AWR) | SMORe (w/o AWR and Expectile loss) |
|------|-------|-----------------|-------------------------------------|
| FetchReach | $35.08\pm_{0.54}$ | $0.30\pm_{0.29}$ | $0.10\pm_{0.13}$ |
| SawyerReach | $36.26\pm_{0.93}$ | $29.31\pm_{0.53}$ | $29.64\pm_{0.62}$ |
| SawyerDoor | $20.28\pm_{2.65}$ | $5.06\pm_{0.52}$ | $2.11\pm_{1.59}$ |
| FetchPick | $26.47\pm_{0.34}$ | $1.79\pm_{0.65}$ | $1.77\pm_{1.51}$ |
| FetchPush | $26.83\pm_{1.21}$ | $4.60\pm_{2.51}$ | $2.69\pm_{1.01}$ |
| FetchSlide | $4.99\pm_{0.40}$ | $0.22\pm_{0.33}$ | $0.50\pm_{0.02}$ |

Table 16: Ablating practical components of SMORe. Without adapting for offline setting we consider in this work by using in-sample maximization or constrained policy optimization using AWR the performance degrades as expected. Without in-sample-maximization value function explodes in the offline setting and using policy that maximizes Q can often select OOD actions leading to poor performance.

