# OpenReview forum: "Score Models for Offline Goal-Conditioned Reinforcement Learning"
_ICLR.cc/2024/Conference — ICLR 2024 poster_

### Official Review · Reviewer_9ANq · 2023-10-22

**Soundness:** 2 fair
**Presentation:** 3 good
**Contribution:** 3 good
**Rating:** 6
**Confidence:** 4

**Summary:**

This paper formulates the offline goal conditioned reinforcement leanring (GCRL) problem as an occupancy matching problem and leverages the popular DICE methods to transform this problem into an objective that can be exclusively trained on offline dataset. Specifically, they adopts ValueDICE[1]-like method for this reformulation. The authors argue that by doing so, the offline GCRL problem can be solved without necessitating a well-trained discriminator. Instead, it learns unnormalized densities or scores that allow it to produce optimal goal-reaching policies, greatly enhancing the training results. In addition, this paper does not directly optimize the objectives derived from the DICE reformulation (Eq. 9), but adopts IQL[2] to optimize some surrogate objectives (Eq. 10-12). From my perspective, this approach mitigates the side-effect of residual learning in Eq. 9 since the surrogate objective in Eq. 11 is optimized via semi-gradient, which is believed to enjoy better performances than residual learning. This paper conducts some experiments on low-dimensional and image-based tasks to support the effectiveness of the proposed method, and also carries out some ablation stuidies on stochastic environments with varied noise levels.

[1] Imitation Learning via Off-Policy Distribution Matching, ICLR 2020

[2] Offline Reinforcement Learning with Implicit Q-Learning, ICLR 2022

**Strengths:**

1. Despite GoFAR[3] already employing DICE methods to address offline GCRL problem, this paper is the first to eliminate the need for an additional discriminator.
2. The experimental results are promising, outperforming baselines (especially the most related GoFAR[3]) across a wide range of evaluation settings.

[3] How far i'll go: offline goal-conditioned reinforcement learning via f-advantage regression, NeurIPS 2022.

**Weaknesses:**

## Problem formulation
1. The target goal-transition distribution $q(s,a,g)$ defined in Section 3.1 might not be a valid discounted visitation distribution that fulfills the Bellman-flow constraint. Therefore, this suggests that the  occupancy matching problem in Eq. 4 may not be appropriately formulated.

## Experiment
2. This paper optimizes surrogate objectives using IQL[2] rather than directly solving the objective derived from DICE reformulation, but does not provide explainations for this choice. Moreover, this paper does not conduct enough ablation studies on this aspect. It raises a question about the performance of GoFAR using this same technique. In my view, optimizing the surrogate objectives in Eq. 10-12 can mitigate the side-effect of residual learning in Eq. 9. Therefore, GoFAR may also obtain large performance gains using the IQL tricks.
3. Given that GoFAR does not employ IQL surrogate objectives, I cannot ensure the comparison in Figure 2 is fair. It would be more fair if the authors could also apply this technique to GoFAR and compare it against GoFAR using this approach.

## Others
4. Some potential overclaims. Section 3.1 is essentially an extension of the conclusion of GoFAR[3] from state-occupancy matching to state-action-occupancy matching. The authors should provide more discussions on the relationships with GoFAR. Although they have made statements like "Proposition 1 extends the insights of formulating GCRL as an imitation learning problem from Ma et al. for goal-transition distributions when matching state-action-goal visitations", this similarity should be made clearer.
5. The Problem Formulation section (Section 2) is largely similar to the one in GoFAR[3] paper, with only minor rewording. In my view, this section should be reorganized or rewording a lot to avoid potential plagiarism.

**Questions:**

Please refer to weakness for details.

---

> ### Author Response · Authors · 2023-11-14
> **Response to Reviewer 9ANq**
>
> We thank the reviewer for their comments and feedback. We provide clarifications below and would be happy to discuss further.
>
> 1. **The target goal-transition distribution defined in Section 3.1 might not be a valid discounted visitation distribution that fulfills the Bellman-flow constraint. Therefore, this suggests that the occupancy matching problem in Eq. 4 may not be appropriately formulated.**
>
> We would like to clarify that irrespective of the goal-transitions distribution being an infeasible visitation distribution (which we discuss in the paper as well), the distribution matching objective in Eq.4 still maximizes a lower bound to the goal-conditioned RL objective. This is proved in Propositions 1 and 2 in the paper.
>
>
>
> 2. **This paper optimizes surrogate objectives using IQL. It raises a question about the performance of GoFAR using this same technique. Therefore, GoFAR may also obtain large performance gains using the IQL tricks.**
>
> We would first like to point out a common misconception: GoFAR does not learn via residual learning even though it seems like it from Algorithm 2 in their paper. Their official code [linked here](https://github.com/JasonMa2016/GoFAR/blob/cfb2d4e36d69c5ec16edf6d127464eb263925e70/rl_modules/gofar_agent.py#L171) actually shows that semi-gradient learning is being used in GoFAR.
>
> Our practical algorithm adapts the derived GCRL objective for the offline setting by using in-sample maximization to avoid overestimation. We appreciate the suggestion and have added this ablation in the Appendix [Table 13]. GoFAR with IQL is seen to have nearly the same performance as vanilla GoFAR.
>
> We note that the in-sample maximization and the constrained policy optimization used in the practical algorithm is not a contribution of our work and is simply taken from [1,2,3] and it has been shown previously unconstrained optimization is detrimental in offline setting [4].
>
>
> 4. **The authors should provide more discussions on the relationships with GoFAR. Proposition 1 similarity should be made clearer.**
>
>
> We appreciate the suggestion and have clarified this in the updated paper. Our contribution lies mainly in proposing a mixture-distribution matching objective for GCRL as well as providing a connection to the original GCRL objective in Proposition 2.
>
>
>  We do want to point out that while Proposition 1 bears similarity in the high-level intuition of distribution matching but differs in the fact that GoFAR matches state-goal transitions, which requires them to construct another lower bound (see Proposition 4.2 and Lemma B.2 in GoFAR[5]) in order to obtain a tractable optimization objective. In contrast, goal-transition distributions allow us to craft an objective that we can directly optimize. We believe that this is an important observation in itself.
>
> Please find our expanded discussion on the theoretical and algorithmic differences of our method compared to GoFAR in the general discussion.
>
> 5. **The Problem Formulation section (Section 2) is largely similar to the one in GoFAR paper, with only minor rewording. In my view, this section should be reorganized**
>
> We appreciate the reviewer for pointing this out. Although our problem setting is exactly the same we recognize the importance of setting up our own context for the work. We have updated the Problem Formulation to address reviewers' concerns. The suggestions have been incorporated into the revision.
>
> ----------
>
> Please let us know if there are any remaining questions or concerns. We hope the reviewer can reassess our work in light of these clarifications and additional empirical results.
>
> ----------
>
> References:
>
> [1]:Kostrikov, Ilya, Ashvin Nair, and Sergey Levine. "Offline reinforcement learning with implicit q-learning." arXiv preprint arXiv:2110.06169 (2021).
>
> [2]: Peng, Xue Bin, et al. "Advantage-weighted regression: Simple and scalable off-policy reinforcement learning." arXiv preprint arXiv:1910.00177 (2019).
>
> [3]: Sikchi, Harshit, et al. "Dual rl: Unification and new methods for reinforcement and imitation learning." Sixteenth European Workshop on Reinforcement Learning. 2023.
>
> [4]: Fujimoto, Scott, David Meger, and Doina Precup. "Off-Policy Deep Reinforcement Learning without Exploration. CoRR abs/1812.02900 (2018)." arXiv preprint arXiv:1812.02900 (2018).
>
> [5]: Ma, Yecheng Jason, et al. "How Far I'll Go: Offline Goal-Conditioned Reinforcement Learning via $ f $-Advantage Regression." arXiv preprint arXiv:2206.03023 (2022).

---

> > ### Comment · Reviewer_9ANq · 2023-11-17
> > **Thank you!!**
> >
> > Dear authors,
> >
> > Thanks for the thorough responses and my concerns are well-resolved. Thus, I'm happy to increase my score to 6. Thank you!!

---

### Official Review · Reviewer_ojhF · 2023-10-24

**Soundness:** 2 fair
**Presentation:** 2 fair
**Contribution:** 2 fair
**Rating:** 6
**Confidence:** 4

**Summary:**

This paper proposes an offline GCRL algorithm from the occupancy matching perspective. The problem setting basically follows GoFAR [1], using a similar DICE-based construction. The difference is that GoFAR is formulated as a V-DICE (learn V and perform goal-conditioned state occupancy matching) and does not use HER; while this paper adopts Q-DICE (learn both Q and $\pi$ by solving a max-min problem, and performs goal-conditioned state-action occupancy matching) and uses HER to generate desirable goal-reaching data ($q(s,a,g)$). The tricky part is that the final practical algorithm is essentially a goal-conditioned version of in-sample learning algorithm which bears lots of similarities with methods like IQL[2], SQL/EQL[3], and XQL[4]. Although such in-sample learning algorithms like SQL/EQL and XQL have been show to have some connection with DICE-based methods, there are some important distinctions. There are some noticeable theoretical gaps here. The proposed method has a number of strange design choices and the algorithm development is not very principled in several places. See the following strengths and weaknesses for detailed comments.

**Strengths:**

- The proposed practical algorithm provides a reasonable approach to combine goal-conditioning and in-sample learning offline RL.
- The performance is good in a number of GCRL tasks. The experiments are comprehensive.
- Provide experiments on vision-based tasks.

**Weaknesses:**

There are several key weaknesses in the paper.
- The problem setting and Section 3.1 largely follow GoFAR[1] with minor changes. A key motivation from the authors is that methods like GOFAR need an unstable discriminator-based construction. However, this does not really hold. As in the GoFAR paper[1], their authors clearly mentioned in the paper that the discriminator can be bypassed by using the reward in the dataset.
- Although using HER to generate augmented goal-transition samples could potentially improve distribution coverage and may contribute to certain level performance improvement. However, as discussed in GoFAR, using HER could also lead to sensitive hyperparameter tuning and suffer from hindsight bias.
- The biggest problem of this paper is the gap between theoretical derivation and the practical algorithm. Based on the augmented samples from HER, the proposed method constructs a goal-conditioned Q-DICE objective which needs to solve a max-$\pi$ and min-$S$ (analogous to Q function in typical Q-DICE algorithm like AlgaeDICE[5]). This actually caused some stability issues due to extracting $\pi$ through the max-min optimization problem. Hence the practical algorithm directly jumps to an in-sample learning framework which is similar to IQL[2], SQL/EQL[3], and XQL[4]. Although there are some connections between the DICE-based method and previous in-sample learning methods, there are also some distinctions. An apparent difference is that the DICE-based method requires minimizing $S$ (analogous to Q in other Q-DICE methods) in the first term of Eq.(9) using samples from initial states $d_0$, while in in-sample learning algorithms, this can be sampled from the whole dataset $\mathcal{D}$ (Eq.(10)). Second, DICE-based method only learn Q (similar to S in this paper) or V, while the previous in-sample learning algorithms learn both Q and V. In my opinion, the paper starts with a DICE formulation and goes a long way to turn it into a non-DICE algorithm. If the authors check the SQL/EQL[3] paper, a more straightforward approach will be starting with its implicit value regularization framework and designing a proper, goal-conditioned regularization function $f$, which will provide a neat and more principled algorithm.
- The proposed algorithm has many hyperparameters, e.g. $\tau$ in Eq.(10), $\beta$ in Eq.(11), and $\alpha$ in Eq.(12). The paper conducts heavy tuning to obtain the best performance. First of all, for an offline RL algorithm, introducing too many hyperparameters and requiring heavy parameter tuning is an extremely bad practice. In practical offline RL applications, it is almost impossible to evaluate or tune model parameters given restricted access to the real environment. In practice, no one will use an offline RL algorithm if it needs careful hyperparameter tuning to achieve good performance.


**References:**

[1] Ma, J. Y., et al. Offline goal-conditioned reinforcement learning via $ f $-advantage regression. NeurIPS 2022.

[2] Kostrikov I, Nair A, Levine S. Offline Reinforcement Learning with Implicit Q-Learning ICLR 2022.

[3] Xu, H., et al. Offline RL with no OOD actions: In-sample learning via implicit value regularization. ICLR 2023.

[4] Garg, D., Hejna, J., Geist, M., & Ermon, S. Extreme Q-Learning: MaxEnt RL without Entropy. ICLR 2023.

[5] Nachum, O., Dai, B., Kostrikov, I., Chow, Y., Li, L., & Schuurmans, D. Algaedice: Policy gradient from arbitrary experience.

**Questions:**

- Please report the hyperparameter $\alpha$ values in your experiments.
- How will the proposed method perform if not tuned individually for each task? Such as using 1~3 sets of hyperparameters.
- All datasets in the experiments are mixed with some expert data. How will the proposed method perform if only uses sub-optimal data samples?

---

> ### Author Response · Authors · 2023-11-14
> **Response to Reviewer ojhF (1/3)**
>
> We thank the reviewer for their comments and feedback. We provide clarifications below and would be happy to discuss further.
>
> 1. **A key motivation from the authors is that methods like GOFAR need an unstable discriminator-based construction. However, this does not really hold. As in the GoFAR paper[1], their authors clearly mentioned in the paper that the discriminator can be bypassed by using the reward in the dataset.**
>
> While GoFAR author suggests that discriminator can be bypassed by using rewards, their experiments are hardcoded to use discriminator (official code line [link](https://github.com/JasonMa2016/GoFAR/blob/cfb2d4e36d69c5ec16edf6d127464eb263925e70/train.py#L130). Please find our results using reward function instead of a discriminator with GoFAR in Table 13 of the revised paper and an extended discussion of differences with GoFAR in the general response comment. GoFAR without discriminator reduces to an RL with a sparse rewards algorithm and has poor performance.
>
>
> 2. **Although using HER to generate augmented goal-transition samples could potentially improve distribution coverage and may contribute to certain level performance improvement. However, as discussed in GoFAR, using HER could also lead to sensitive hyperparameter tuning and suffer from hindsight bias.**
>
> Figure 4 in GoFAR demonstrates that HER does not improve performance for GoFAR. Our work, as well as GoFAR, compares to baselines with access to HER data augmentation. A fair comparison directs that we hold our method to the same standards. Removing hindsight bias from HER is an orthogonal exploration (ex. [1]) and our method presents a goal-conditioned policy learning approach applicable for any offline data distribution. We have added a comparison of our method without HER in Table 12 where it can be seen that HER contributes to a small performance gain, even without which we outperform GoFAR.
>
> 3. **The final practical algorithm is essentially a goal-conditioned version of in-sample learning algorithm which bears lots of similarities with methods like IQL, SQL/EQL, and XQL.**
>
>
> The work diverges from prior works like  IQL, SQL/EQL, and XQL in the fundamental sense that we optimize for \textit{distribution matching as opposed to reward maximization}. A goal-conditioned version of an in-sample learning algorithm would consider minimizing the bellman optimality error with sparse reward functions, whereas our loss function presents a reward-free contrastive objective with smoothness regularization.
>
> Our practical objective uses in-sample maximization as a tool rather than the algorithm. In-sample maximization is a general technique to avoid overestimation of the maximum by avoiding OOD data points and is not specific to offline RL methods [2]. We also note that a connection to in-sample learning methods should not be surprising as prior work [3] shows in-sample learning can be derived from DICE.

---

> > ### Author Response · Authors · 2023-11-14
> > **Response to Reviewer ojhF (2/3)**
> >
> > 4. **The biggest problem of this paper is the gap between theoretical derivation and the practical algorithm. The practical algorithm directly jumps to an in-sample learning framework which is similar to IQL, SQL/EQL, and XQL. DICE algorithms are different from in-sample maximization algorithms because a. their initial state distribution is different from whole dataset b. they only learn Q or V, not both**
> >
> > We respectfully disagree and demonstrate below how our practical algorithm closely follows our theoretical derivation and adapts to the offline setting we consider. Our derived objective maximizes over policies $\pi$ and minimizes over $S$. The objective contains terms that depend on next state making it susceptible to overestimation if $\pi$ chooses OOD actions when it is maximized at any given state. Hence, in our practical algorithm, we use:
> >
> > $$
> > \text{in-sample-max}\_{\pi}[g(\pi)] ~~ \text{as a surrogate to} ~~ \max\_{\pi}[g(\pi)].
> > $$
> >
> > In-sample maximization is a mathematical tool to avoid overestimation that has been applied to work outside offline RL methods (eg. [2,8]).
> >
> > Second, the reviewer mentions that DICE necessitates using samples from initial state distribution. This is incorrect. In a number of prior DICE methods, it has been found that setting the initial state distribution to the entire replay-buffer/offline-dataset distribution leads to more stable and performant optimization [3,4]. [3] connects the link between using the offline dataset as the initial state distribution directly to in-sample maximization. Finally, using a wider initial state distribution of offline dataset does not affect the policy performance at the environment's initial state distribution as long as the function-approximator has enough capacity (in our case they are neural networks) since the policy will only learn to achieve near-optimal behaviors from a wider variety of states.
> >
> > Third, we respectfully disagree that DICE methods cannot use both Q and V for learning. This has been done in the past in OptiDICE[9, ICML 21] as well as [3]. As opposed to using single sample targets, learning Q reduces variance in targets by averaging over stochasticity in dynamics.
> >
> > We appreciate the reviewer's suggestion to make the connection to SQL/EQL, but point out that one key difference is that SQL/EQL are reward maximization algorithms with policy regularization whereas we aim to derive a distribution matching algorithm using policy visitations. For this purpose, we believe the duality in RL approach by Nachum et al[5] is more suitable. We have added a comparison with in-sample learning methods in Table 15 where we find SQL/EQL/XQL to be particularly more unstable in sparse reward settings.
> >
> >  5. **The proposed algorithm has many hyperparameters, e.g.  $\tau$ in  Eq.(10), $\beta$ in Eq.(11), and $\alpha$ in Eq.(12) ..and requires many hyperparameter tuning.**
> >
> > We believe there is a misunderstanding about the hyperparameter tuning done in this work. We clarify that $\alpha$ and $\beta$ were fixed to 3.0 and 0.5 throughout all the environments. The only parameter tuned per set of environments is $\beta$ which controls the conservativeness of the value function and is an MDP-dependent parameter. As we show in the general discussion comment, we found very minimal hyperparameter tuning was required and the final results use a fixed value of $\beta$ per class of environments. We have added this discussion in the appendix. Further, Table 9 demonstrates our method is robust across a range of $\beta$ values. This is significantly less hyperparameters tuned compared to popular algorithms like CQL[6] (pessimism, regularizer, SAC parameters), IQL[7] (expectile, policy temperature).
> >
> > **Question 1: Please report the hyperparameter values in your experiments.**
> >
> > We have added this in the updated appendix.
> >
> > **Question 2: How will the proposed method perform if not tuned individually for each task? Such as using 1~3 sets of hyperparameters.**
> >
> > As we discussed above our method already uses one set of hyperparameters per environment class. We have clarified this in the appendix now.
> >
> > **Question 3: All datasets in the experiments are mixed with some expert data. How will the proposed method perform if only uses sub-optimal data samples?**
> >
> > We appreciate the reviewer's suggestion and have added this to Appendix (Table 14). In most environments, random data has no examples of methods reaching goal successfully and in turn, all methods are not expected to perform well.
> >
> > ----------
> >
> > We hope the reviewer can consider these clarifications and additional empirical results when assessing our work. Please let us know if there are further questions or concerns.

---

> > > ### Author Response · Authors · 2023-11-14
> > > **Response to Reviewer ojhF (3/3)**
> > >
> > > References:
> > >
> > > [1] Schramm, Liam, et al. "USHER: Unbiased Sampling for Hindsight Experience Replay." Conference on Robot Learning. PMLR, 2023.
> > >
> > > [2] Hejna, Joey, and Dorsa Sadigh. "Inverse Preference Learning: Preference-based RL without a Reward Function." arXiv preprint arXiv:2305.15363 (2023).
> > >
> > > [3] Sikchi, Harshit, et al. "Dual rl: Unification and new methods for reinforcement and imitation learning." Sixteenth European Workshop on Reinforcement Learning. 2023.
> > > [4] Kostrikov, Ilya, Ofir Nachum, and Jonathan Tompson. "Imitation learning via off-policy distribution matching." arXiv preprint arXiv:1912.05032 (2019).
> > >
> > > [5] Nachum, Ofir, and Bo Dai. "Reinforcement learning via fenchel-rockafellar duality." arXiv preprint arXiv:2001.01866 (2020).
> > >
> > > [6] Kumar, Aviral, et al. "Conservative q-learning for offline reinforcement learning." Advances in Neural Information Processing Systems 33 (2020): 1179-1191.
> > >
> > > [7] Kostrikov, Ilya, Ashvin Nair, and Sergey Levine. "Offline reinforcement learning with implicit q-learning." arXiv preprint arXiv:2110.06169 (2021).
> > >
> > > [8] Hansen, Alex. "The three extreme value distributions: An introductory review." Frontiers in Physics 8 (2020): 604053.
> > >
> > > [9] Lee, Jongmin, et al. "Optidice: Offline policy optimization via stationary distribution correction estimation." International Conference on Machine Learning. PMLR, 2021.

---

> > > > ### Comment · Reviewer_ojhF · 2023-11-15
> > > > **Thanks for the response**
> > > >
> > > > I'd like to thank the authors for the detailed responses and clarifications. While I still have some reservations regarding the gaps between the theoretical derivation and the final practical algorithm, I think the authors have done a reasonable job in their rebuttal discussions. Therefore, I have increased my evaluation score.

---

### Official Review · Reviewer_V1Bw · 2023-10-28

**Soundness:** 3 good
**Presentation:** 2 fair
**Contribution:** 3 good
**Rating:** 6
**Confidence:** 3

**Summary:**

This paper proposes a method for offline goal-conditioned RL, integrating occupancy matching with a convex dual formulation so that the learning objective is converted for better leveraging on suboptimal offline data. Instead of behavior cloning, contrastive RL, or RL with sparse reward, the proposed method is built upon the direction of occupancy matching but without learning an additional discriminator. The proposed method is supported by theorems and evaluated with comprehensive simulation experiments.

**Strengths:**

* The paper is well-written, and the concepts are clearly and concisely explained
* The authors provide theoretical contributions.
* The paper's contributions are supported by empirical analyses on a range of benchmarks, demonstrating the advantage of using SMORe for suboptimal offline data, especially the evaluation of robustness and high-dimensional observation space.

**Weaknesses:**

* Could you please elaborate more about the technical differences between SMORe and GoFAR? My understanding is that the main difference is whether the training involved a discriminator. Does any other difference in details improve the novelty of SMORe?

**Questions:**

* Is the "0.25" in equation 9 a fixed number or a kind of coefficient that can be tuned?

* Any reason why you make the mixture of random/medium and expert data following 4.1 EXPERIMENTAL SETUP?  I thought that there is already a pipeline for collecting random, medium, and expert sub-dataset?

* Can you explain why SMORe has an even higher discounted return in Figure 2 under 0.5 noise level compared with 0 noise? In addition, I am interested in the variance of Figure 2.

* Is there any insight or analysis into why pick and place tasks in Figure 3 are relatively difficult for other baselines compared with the remaining tasks while the proposed method can have a significant improvement in pick and place tasks?

---

> ### Author Response · Authors · 2023-11-14
> **Response to Reviewer V1Bw**
>
> We thank the reviewer for their comments and feedback. We provide clarifications below and would be happy to discuss further.
>
> 1. **Could you please elaborate more about the technical differences between SMORe and GoFAR? My understanding is that the main difference is whether the training involved a discriminator. Does any other difference in details improve the novelty of SMORe?**
>
> Our novelty compared to GoFAR can be summarized in two parts: a) Theoretical: 1. We present a novel objective of mixture distribution matching for GCRL that provably optimizes the GCRL objective without requiring a discriminator. 2.
> GoFAR considers matching state-goal distributions which requires constructing a loose lower bound (Proposition B.1 and Lemma B.1 in [1]) that is not tight for any $f$-divergence. Using goal-transition distribution (state-action-goal) allows us to create a lower bound to the original GCRL objective that is tight for KL divergence. b) Algorithmic: 1. GoFAR requires a three-step learning process of first learning discriminator, then value, and finally extracting a policy. Our method presents a simpler two-step approach by directly learning the optimal score function and corresponding policy. Empirically our method achieves significant gains across the board compared to GoFAR.
>
> We refer the reviewer to the general discussion comment where we present a more thorough comparison with GoFAR.
>
>
> 2. **Is the "0.25" in equation 9 a fixed number or a kind of coefficient that can be tuned?**
>
> 0.25 is a fixed number dictated by the use of $\chi^2$ divergence. It weighs the smoothness regularization vs the contrastive-objective ensuring the resulting objective we optimize has principled connections to the original GCRL objective we are interested in.
>
> 3. **Any reason why you make the mixture of random/medium and expert data following 4.1 EXPERIMENTAL SETUP? I thought that there is already a pipeline for collecting random, medium, and expert sub-dataset?**
>
> We do not collect our own dataset for Fetch or Sawyer environments. These datasets are part of the benchmark established by previous work [1,2]. We explain the dataset collection procedure used the by previous work in the appendix only for completeness.
>
> Note that the locomotion tasks for GCRL as well their datasets are introduced by us for the first time.
>
> 3.**Can you explain why SMORe has an even higher discounted return in Figure 2 under 0.5 noise level compared with 0 noise? In addition, I am interested in the variance of Figure 2.**
>
> We suspect that is a consequence of different quality of transitions in dataset collected with different noise levels. It can also be a result of noise with finite seeds. We appreciate the suggestion of showing variance and have included the standard deviation now in the updated paper.
>
> 4. **Is there any insight or analysis into why pick and place tasks in Figure 3 are relatively difficult for other baselines compared with the remaining tasks while the proposed method can have a significant improvement in pick and place tasks?**
>
> The final pick and place task is more complicated in the sense that it requires precise manipulation. The gripper can collide with the drawer resulting in failure to complete the tasks. Our hypothesis is that, in other tasks, mistakes are more forgivable and allow other baselines to perform reasonably as well.
>
> --------------------
>
> Please let us know if there are any remaining questions or concerns. We hope the reviewer can reassess our work in light of these clarifications and additional empirical results.
>
> -----------------
>
> References:
>
> [1]  Ma, Yecheng Jason, et al. "How Far I'll Go: Offline Goal-Conditioned Reinforcement Learning via $ f $-Advantage Regression." arXiv preprint arXiv:2206.03023 (2022).
>
> [2] Yang, Rui, et al. "Rethinking goal-conditioned supervised learning and its connection to offline rl." arXiv preprint arXiv:2202.04478 (2022).

---

> ### Author Response · Authors · 2023-11-18
> **Follow up on rebuttal**
>
> We thank the reviewer for taking the time to provide a thoughtful review and are motivated that they found the work well-written and acknowledged our theoretical contributions and empirical experiments.
>
> We believe we have responded to the questions and concerns in full, but if something is missing please let us know and we would be happy to add it. If we have addressed your concerns we would appreciate it if the reviewer could reassess our work in light of these clarifications and additional empirical results.

---

> > ### Comment · Reviewer_V1Bw · 2023-11-22
> >
> > Thank you for all your detailed clarifications, which make this manuscript much clearer. I do not have any further questions, and I will keep my original score.

---

### Official Review · Reviewer_Wf6E · 2023-10-31

**Soundness:** 3 good
**Presentation:** 3 good
**Contribution:** 2 fair
**Rating:** 6
**Confidence:** 3

**Summary:**

The paper introduces a novel approach for offline goal-conditioned reinforcement learning, known as SMORe, which is derived from a mixture-distribution matching perspective and eliminates the need for learning a discriminator. The paper demonstrates that SMORe outperforms state-of-the-art baselines in various robot manipulation and locomotion tasks, including high-dimensional observations.

**Strengths:**

The paper is well-written and clearly motivates the problem of offline GCRL. It uses convex duality theory to derive a dual optimization problem that can use offline data to learn score functions and policies, which provides a rigorous theoretical analysis of the proposed method. It also presents extensive empirical results that demonstrate the effectiveness and robustness of SMORe on challenging benchmarks.

**Weaknesses:**

One weakness of the paper is that the claim of being discriminator-free is somewhat overclaiming. From Eq. 12, the S-function can be seen as a Q-function and the M-function can be seen as a V-function. In this case, although the framework does not have an explicit discriminator, the S-function actually plays the role of a discriminator. What's more, the proposed method requires two networks, S and M, to learn the optimal policy, while many works only require a network (such as contrastive RL). This implies that the proposed method has more parameters and computational complexity.
Another weakness is that it is unclear how much of the performance gain comes from each component. A possible suggestion is to add an experiment without using expectile regression and AWR, as well as experiments that show how they work separately. This would clarify the role of each part in the proposed method.

**Questions:**

Why do the WGCSL and GCSL methods have similar performance in Table 7-10, while show a large difference in Table 1 for the following four environments: CheetahTgtVel-m-e, CheetahTgtVel-r-e, AntTgtVel-m-e and AntTgtVel-r-e? What factors could explain this discrepancy?

---

> ### Author Response · Authors · 2023-11-14
> **Response to Reviewer Wf6E**
>
> We thank the reviewer for their comments and feedback. We provide clarifications below and would be happy to discuss further.
>
> 1. **One weakness of the paper is that the claim of being discriminator-free is somewhat overclaiming. In this case, although the framework does not have an explicit discriminator, the S-function actually plays the role of a discriminator.**
>
>
> We clarify that S-function is not a discriminator, as it reasons about the unnormalized long-term value of an action in reaching a goal. Contrary to GoFAR which requires a three-step process - a. Learn reward as discriminator b. Train optimal value function using this reward c. Extract Policy using the value function, our method requires two steps -- Learn the optimal score/value function and extract policy. Our score function is not learned via a classification loss and does not resemble a discriminator.
>
>
>
> 2. **What's more, the proposed method requires two networks, S and M, to learn the optimal policy, while many works only require a network (such as contrastive RL). This implies that the proposed method has more parameters and computational complexity.**
>
> While works like contrastive RL only require a single network, they are provable suboptimal as we discuss in the paper. While our method is indeed more computationally complex than contrastive RL, it bears the same computational complexity as GoFar[1] and IQL[2]. Additionally, it is stable to train with little hyperparameter tuning ($\tau$) in all environments and also robust to few-expert coverage in the offline dataset. In real-world problems, often data not the compute is often the bottleneck.
>
>
>
> 3.  **Another weakness is that it is unclear how much of the performance gain comes from each component. A possible suggestion is to add an experiment without using expectile regression and AWR, as well as experiments that show how they work separately. This would clarify the role of each part in the proposed method.**
>
>
>
> Our practical algorithm adapts the derived GCRL objective for the offline setting by using in-sample maximization to avoid overestimation. We appreciate the suggestion and have added this ablation in the Appendix Table 16.  We observed a value function blowup leading to poor policy performance without using techniques to constrain learned policy to the offline data distribution. An identical approach was taken by [6, NeurIPS 2023] when adapting their preference learning method to an offline setting.  We have also added another ablation of using expectile loss and AWR in GoFAR in Table 13.
>
>
> We note that the in-sample maximization and the policy optimization used in the practical algorithm is not a contribution of our work and is taken from [2,3,4] and it has been shown previously unconstrained optimization is detrimental in offline setting [5].
>
>
> 4. **Why do the WGCSL and GCSL methods have similar performance in Table 7-10, while show a large difference in Table 1 for the following four environments: CheetahTgtVel-m-e, CheetahTgtVel-r-e, AntTgtVel-m-e and AntTgtVel-r-e? What factors could explain this discrepancy?**
>
> We appreciate the reviewer pointing out this interesting observation for a comparison of baselines between WGCSL and GCSL. Our hypothesis was the following: The datasets we consider are bi-modal consisting of transitions coming from suboptimal trajectories and expert trajectories. GCSL given a goal, is able to easily distinguish goals to select a particular mode and get good performance. WGCSL relies on learning a critic that additionally influences the policy. Any error in the critic, can translate to policy error which compounds over time.
>
> -------------------
>
> Please let us know if there are any remaining questions or concerns. We hope the reviewer can reassess our work in light of these clarifications and additional empirical results.
>
> ----------------
>
> References:
>
>
> [1] Ma, Yecheng Jason, et al. "How Far I'll Go: Offline Goal-Conditioned Reinforcement Learning via $ f $-Advantage Regression." arXiv preprint arXiv:2206.03023 (2022).
>
> [2] Kostrikov, Ilya, Ashvin Nair, and Sergey Levine. "Offline reinforcement learning with implicit q-learning." arXiv preprint arXiv:2110.06169 (2021).
>
> [3] Peng, Xue Bin, et al. "Advantage-weighted regression: Simple and scalable off-policy reinforcement learning." arXiv preprint arXiv:1910.00177 (2019).
>
> [4] Sikchi, Harshit, et al. "Dual rl: Unification and new methods for reinforcement and imitation learning." Sixteenth European Workshop on Reinforcement Learning. 2023.
>
> [5] Fujimoto, Scott, David Meger, and Doina Precup. "Off-Policy Deep Reinforcement Learning without Exploration. CoRR abs/1812.02900 (2018)." arXiv preprint arXiv:1812.02900 (2018).
>
> [6]: Hejna, Joey, and Dorsa Sadigh. "Inverse Preference Learning: Preference-based RL without a Reward Function." arXiv preprint arXiv:2305.15363 (2023).

---

> > ### Author Response · Authors · 2023-11-18
> > **Follow up on rebuttal**
> >
> > We thank the reviewer for taking the time to provide a thoughtful review and are motivated that they found the work well-written with rigorous theoretical analysis and extensive empirical experiments.
> >
> > We believe we have responded to the questions and concerns in full, but if something is missing please let us know and we would be happy to add it. If we have addressed your concerns we would appreciate it if the reviewer could reassess our work in light of these clarifications and additional empirical results.

---

### Author Response · Authors · 2023-11-14
**General Response**

We would like to thank the reviewers for their detailed comments. We respond to the individual reviews below. We’ve also updated the paper (changes in red) with the following modifications to address reviewer's suggestions and concerns:

[Main paper] a) Updated problem statement b. Updated result highlighting with a better statistical test c. Improved presentation

[Appendix] [a] Added details on hyperparameters [b] Added intuition for proposition 2. [c] Added ablation experiments on in-sample learning, HER, offline components in the practical algorithm, using 0 expert data, and GoFAR with sparse reward.

## How does our work differ from GoFAR?

We tackle the same problem setting as GoFAR in our work. Our contribution is a new approach for offline GCRL derived by proposing a  mixture-distribution matching objective that leads to discriminator-free training. We provide a simpler approach to distribution matching by removing the need for policy learning over a learned component(discriminator as a reward) and instead directly performing policy learning. Our method improves learning performance across a wide range of tasks compared to GoFAR. We elaborate on the differences below.

GoFAR's objective is based on distribution matching between current policy visitation and an infeasible distribution will defined on state-goal tuple ($p(s,g)\propto e^{r(s,g)}$):

$$
    D_f(d^{\pi_g}(s,g) \| p(s,g))
$$
To make this optimization tractable, they consider optimizing a lower bound:
\begin{equation}
    - D\_{KL}(d^{\pi\_g}(s,g) \| p(s,g)) \ge \mathbb{E}\_{d^{\pi}}[{\log \frac{p(s,g)}{d^O(s,g)}}] - D\_f(d^{\pi_g}(s,a,g) || d^{O}(s,a,g))
\end{equation}
The above conversion suffers from the following issues:

- The first term is a density ratio that is learned via a discriminator and later used as reward for policy optimization. Logarithm of density ratio is ill-defined whenever the goal distribution does not cover offline distribution ($p(s,g)=0, d^O(s,g)\neq0$). This is likely to hold as offline data will have more support than goal distribution in practice making the method brittle. Furthermore, the policy optimization in MDP suffer from cascading error of $\frac{\epsilon}{1-\gamma}$ for an error of $\epsilon$ in the reward function. Thus if an erroneous discriminator is used as a reward function it will lead to unavoidable cascading errors.
- The lower bound in the second equation above is not tight for any divergence as it contains an unoptimizable term( $D_{KL}(\pi(a|s,g)||\pi^O(a|s,g)$, see Lemma B.2 and Proposition 4.2 in Ma et al[1]), where $\pi^O$ is the policy that generated the offline dataset.
- Although GoFAR claims that the discriminator can be replaced by 0-1 reward function, in practice they obtain performant results using a learned discriminator (see Table 13 in our revised paper for comparison). Using 0-1 reward in GoFAR converts the method from distribution matching to RL with sparse rewards (Eq 9 in Ma et al[1] similar to Nachum et al.[2]). *Our work is able to optimize a distribution-matching objective without a discriminator.*

Our work proposes a way to resolve the above issues of GoFAR by first formulating distribution matching between goal-transition distributions (state-action-goals instead of state-goals). This allows us to bypass creating a loose lower bound. Then we present a distribution-matching objective that removes the need to learn a discriminator.
\begin{equation}
   \min\_{\pi\_g} D_f(\texttt{Mix}\_\beta(d^{\pi\_g}, \rho)  ||  \texttt{Mix}\_\beta(q, \rho) ),
\end{equation}
where $q(s,a,g)$ denotes goal-transition distribution. The mixture of offline data $\rho$ for both distributions being matched is key to learning completely off-policy without relying on a discriminator. We show in Proposition 2, how this leads to a principled GCRL objective.

We summarize key properties and contributions of our new proposed objective for GCRL:

- **A simpler approach**: Directly learns the optimal policy and score as opposed to learning a discriminator, then training value function and policy.
- **GoFAR is not performant without discriminator**: Although GoFAR motivates that discriminator learning is optional, the results present a different picture. Without discriminator learning, GoFAR reduces to a 0-1 sparse reward RL method and as our added experiments in Table 13 of Appendix shows, has poor performance with a sparse reward.
- **Robust to fewer expert coverage in offline data**: Our proposed approach is robust to decreasing expert coverage in the offline dataset and does not require $p(s,g)>0$ whenever  $d^O(s,g)>0$ as GoFar does in the equation shown above. The limited amount of high-quality data in the offline dataset is often a bottleneck.
- **Strong empirical performance**: Our results demonstrate across-the-board improvements obtained by SMORe compared to GoFar.

---

> ### Author Response · Authors · 2023-11-14
> **General Response (2/2)**
>
> References:
>
> 1. Ma, Yecheng Jason, et al. "How Far I'll Go: Offline Goal-Conditioned Reinforcement Learning via $ f $-Advantage Regression." arXiv preprint arXiv:2206.03023 (2022).
>
>
> 2. Nachum, Ofir, and Bo Dai. "Reinforcement learning via fenchel-rockafellar duality." arXiv preprint arXiv:2001.01866 (2020).

---

### Meta-Review · Area_Chair_YmRL · 2023-12-07

**Metareview:**

(a) Summary
The paper introduces SMORe, a novel method for Offline Goal-Conditioned Reinforcement Learning, focusing on learning from offline datasets without custom reward functions. Traditional occupancy matching GCRL methods require training a separate discriminator and policy, which can lead to cascading errors. SMORe looks at a mixture-distribution matching approach, and derives a loss that does not require a separate discriminator. The approach shows notable performance improvements over existing methods in robot manipulation and locomotion tasks, including high-dimensional observation spaces.

(b) Strengths:
(+) Theoretical Contribution and Empirical Validation (Reviewer 9ANq, Reviewer V1Bw, Reviewer Wf6E): The paper introduces a new theoretical framework for GCRL, integrating occupancy matching with a convex dual formulation. This framework is empirically validated across various benchmarks, demonstrating effectiveness in both low and high-dimensional observation spaces.

(+) Algorithmic Novelty (Reviewer 9ANq, Reviewer V1Bw): SMORe distinguishes itself from existing methods like GoFAR by eliminating the need for a discriminator. This is achieved through a reformulation method that follows the principles of ValueDICE[1], in contrast to GoFAR's reliance on SMODICE[2], leading to distinct learning objectives and simplifying the learning process.

(+) Soundness and Clarity of Presentation (Reviewer Wf6E): The paper's approach is grounded in convex duality theory, leading to a dual optimization problem that leverages offline data for learning score functions and policies. The empirical results presented are extensive and support the method's effectiveness.

(c) Weaknesses:

(-) Distinctiveness from GoFAR (Reviewer 9ANq, Reviewer ojhF): The paper's approach, while novel, is not distinctly different from GoFAR in terms of the foundational concept of occupancy matching. GoFAR is formulated as a state-occupancy matching problem and is credited with first identifying that occupancy matching can address the Goal-Conditioned RL problem. In contrast, SMORe is based on a state-action-occupancy matching approach. This similarity leads to questions about the distinctiveness and novelty of SMORe compared to GoFAR, as the core concept of occupancy matching was already introduced by GoFAR.

(-) Gaps between Theoretical Derivation and Practical Algorithm (Reviewer ojhF): There seems to be a gap between the theoretical model and its practical application in SMORe. The algorithm in practice diverges from the original theoretical propositions, which may impact the method's principled development and consistency.

(-) Hyperparameter Sensitivity (Reviewer ojhF): SMORe's performance is potentially limited by its dependency on multiple hyperparameters and the need for extensive tuning, which could limit its practical use in real-world offline RL applications where such tuning is impractical.

**Justification For Why Not Higher Score:**

The decision to accept with poster, rather than spotlight or oral, is because of the limited novelty. The paper's approach, though novel in certain aspects like its discriminator-free method, shares conceptual similarities with previous works, particularly GoFAR. The discussion around the distinctiveness of SMORe suggests that while the paper is an important contribution to the field, it may not represent a significant shift that typically warrants a higher presentation category like an oral or spotlight session.

**Justification For Why Not Lower Score:**

The decision to not reject is because of:
1. Significant theoretical contributions: It advances the understanding of occupancy matching in GCRL by proposing a mixture-distribution matching approach that eliminates the need for a discriminator. These theoretical insights are valuable to the field.
2. Empirical validation and soundness: The paper provides a rigorous empirical evaluation of the proposed method. The extensive experiments demonstrate the effectiveness of SMORe across a range of benchmarks, including complex robot manipulation and locomotion tasks.
3.  Addressing Reviewer Concerns: The authors have made a commendable effort to address the concerns raised by the reviewers. They provided additional clarifications and empirical results in their rebuttal.

---

### Decision · Program_Chairs · 2024-01-16

Accept (poster)